# AutoGUI: Scaling GUI Grounding with Automatic Functionality Annotations from LLMs

## Abstract

User interface understanding with vision-language models has received much attention due to its potential for enabling next-generation software automation. However, existing UI datasets either only provide large-scale context-free element annotations or contextualized functional descriptions for elements at a much smaller scale. In this work, we propose the AutoGUI pipeline for automatically annotating UI elements with detailed functionality descriptions at scale. Specifically, we leverage large language models (LLMs) to infer element functionality by comparing the UI content changes before and after simulated interactions with specific UI elements. To improve annotation quality, we propose LLM-aided rejection and verification, eliminating invalid and incorrect annotations without human labor. We construct an AutoGUI-704k dataset using the proposed pipeline, featuring multi-resolution, multi-device screenshots, diverse data domains, and detailed functionality annotations that have never been provided by previous datasets. Human evaluation shows that the AutoGUI pipeline achieves annotation correctness comparable to trained human annotators. Extensive experimental results show that our AutoGUI-704k dataset remarkably enhances VLM's UI grounding capabilities, exhibits significant scaling effects, and outperforms existing web pre-training data types. We envision AutoGUI as a scalable pipeline for generating massive data to build GUI-oriented VLMs. AutoGUI dataset can be viewed at this anonymous URL: `https://huggingface.co/AutoGUI`.

## 1 Introduction

User interface understanding with visual language models(VLMs) (Hong et al., 2023; Cheng et al., 2024; You et al., 2024a; Lee et al., 2023; Baechler et al., 2024) has received wide attention due to its potential in fundamentally transforming how we interact with software as well as unleashing unseen flexibility for existing apps (Fig. 1). *Functionality prediction*, which aims to understand the semantic purpose and interactive affordance of individual UI elements, is a crucial task that goes beyond previous UI understanding tasks focusing on structural mapping between UI code and visual layout, such as UI REG/REC (Hong et al., 2023; Li et al., 2020a) and diagram to code (Xia et al., 2024; Liu et al., 2023a).

To enhance the UI understanding capability of VLMs, large-scale high-quality training data is indispensable. However, the scale of existing open-source datasets (Li et al., 2020a; Deka et al., 2017a; Li et al., 2020b; Kapoor et al., 2024; Wang et al., 2021) for UI understanding remains on the order of millions, significantly fewer than natural image datasets such as LAION-5B (Schuhmann et al., 2022). Additionally, the prevailing methods (Deka et al., 2017a; Li et al., 2020a) for collecting UI annotation are labor-intensive, leading to prohibitive costs that hinder scalability. Moreover, existing UI understanding datasets predominantly focus on describing either the visual appearance (Li et al., 2020a;b) (e.g., a button beside the navigation bar), element categories (Cheng et al., 2024) (e.g., "menu button"), or brief functions weakly related to the UI context (Bai et al., 2021) (e.g., "show more information") shown in Fig. 2. These datasets lack contextual functional descriptions of UI elements, which poses a challenge for VLMs in comprehending the roles these elements serve within specific UI contexts, such as distinguishing between two visually similar magnifying glass icons that may represent distinct functionalities like searching and zooming.

Figure 1: UI understanding VLMs could plan a trip to the airport by integrating information across different apps and modes of transportation.

To address the challenge. we propose AutoGUI, a scalable and automatic UI data annotation pipeline that provides unlimited UI element functionality annotations. Our annotation pipeline automatically collects UI interaction trajectories and leverages large language models (LLMs) to infer element functionalities based on UI content changes, eliminating the need for manual annotation by human experts. Initially, the proposed pipeline crawls a multitude of interaction trajectories on either a web browser or an Android emulator and captures screenshots at various aspect ratios. Subsequently, we use open-source LLMs (AI@Meta, 2024) to annotate the functionalities of elements on collected GUIs based on changes to UI contents when interacting with these elements. To ensure data quality, LLM-aided rejection is utilized to eliminate invalid samples, such as incompletely rendered UIs. Additionally, inspired by recent works on LLM verification (Weng et al., 2022; Lightman et al., 2023), multiple LLMs are prompted as verifiers to identify false functionality predictions. With both the rejection and verification processes, our pipeline removes unclear and invalid samples. We curate the AutoGUI-704k dataset with the proposed pipeline. AutoGUI-704k contains 704k high-quality functionality grounding and referring tasks used to finetune and evaluate open-source VLMs. With the vast knowledge embedded within LLMs (e.g., Llama-3-70B (AI@Meta, 2024)) and fast inference infrastructure (Kwon et al., 2023; Gugger et al., 2022), our pipeline can efficiently annotate high-quality samples at a large scale and substantially reduced cost compared to traditional methods. Moreover, pioneer experiments find that our pipeline achieves annotation accuracy of **96.7%** comparable to a trained human annotator.

Based on the collected AutoGUI-704k dataset, we finetune open-source VLMs that own little UI grounding capabilities. Experimental results demonstrate that data collected through our AutoGUI pipeline significantly enhances the VLMs' UI grounding accuracy and exhibits remarkable scaling effects. The results also show that our functionality annotation type is superior to the data type directly derived from web HTML code (Hong et al., 2023; Cheng et al., 2024), serving as a promising data source for building VLMs capable of UI grounding.

## 2 RELATED WORKS

### 2.1 RECENT ADVANCEMENT OF VLMS

Recently, a new wave of research has started to enhance LLMs with the capability of processing both visual and textual information (Alayrac et al., 2022; Chen et al., 2023a; Li et al., 2023; Lin et al., 2023a; Liu et al., 2023b; Lin et al., 2023b; Chen et al., 2023b; Lu et al., 2024; Bai et al., 2023; Wang et al., 2024a; Zhu et al., 2024; Wang et al., 2024b; Li et al., 2024; Zhang et al., 2024a; You et al., 2024a; Laurençon et al., 2024; Peng et al., 2024; Driess et al., 2023), opening the new field of Vision Language Model (VLM). Pioneering efforts Flamingo (Alayrac et al., 2022) uses interleaved visual and language inputs as prompts and shows remarkable few-shot visual question-answering capability. Fueled by GPT-4 (Team, 2024), both academia and industry have endeavored to democratize its amazing multimodal reasoning capability. LLaVA (Liu et al., 2023b) and LLaMA-Adapter (Zhang et al., 2024a) have attempted to align vision encoders (Dosovitskiy et al., 2021) with LLMs to

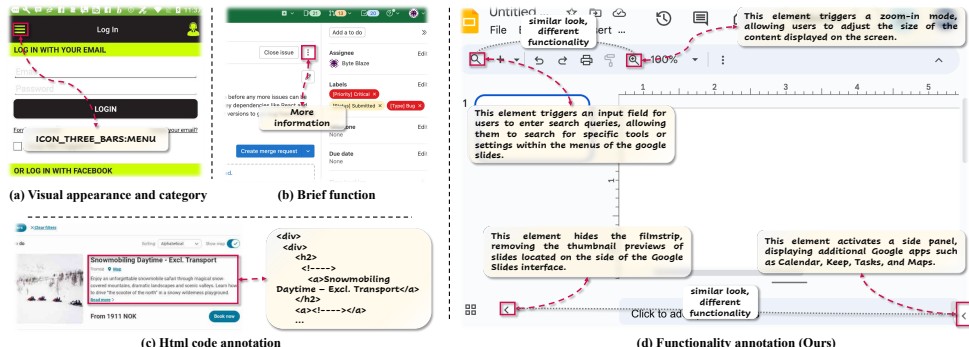

**Figure 2:** Our functionality annotations vs. the annotations provided by existing UI datasets. The proposed AutoGUI (right) can generate element annotations rich in functional semantics.

enable visual instruction following. Models such as VisionLLM (Wang et al., 2024b), Ferret (You et al., 2024a), and Qwen-VL (Bai et al., 2023) further enhance these capabilities with robust visual grounding. Additionally, Research is also expanding into VLM applications in scenarios rich in textual imagery (Tang et al., 2022; Ye et al., 2023b;a; Liu et al., 2024c) and embodied interactions (Driess et al., 2023; Mu et al., 2023), offering new possibilities in multimodal reasoning. Despite these advancements, the domain of UI understanding remains under-explored due to data scarcity. This paper proposes an autonomous UI annotation pipeline to tackle this challenge, aiming to expand the data available for training VLMs in this crucial area.

### 2.2 EXISTING UI DATASETS AND BENCHMARKS

Unlike mature natural image datasets (Russakovsky et al., 2014; Schuhmann et al., 2022), UI understanding datasets have received less attention in computer vision. Several efforts have been made to develop mobile UI modeling datasets (Wang et al., 2021; Li et al., 2020a;b; Bai et al., 2021; Burns et al., 2022), primarily annotating the RICO dataset (Deka et al., 2017b), which includes 72K screenshots from Android apps. Examples include Widget Captioning (Li et al., 2020a), which analyzes captions and linguistic features of UI elements, and RICOSCA (Li et al., 2020b), which maps single-step instructions to UI locations. Recently, MoTIF (Burns et al., 2022) and AITW (Rawles et al., 2023) have been proposed to focus on interpreting high-level instructions in Android environments. However, these manually curated and crowd-annotated datasets are limited in size and costly to update, presenting challenges in adapting to new UI types.

The web scenario has also gained much attention. WebShop (Yao et al., 2022), as an early attempt, introduces a simplified simulator for web navigation tasks. More recent projects, such as Mind2Web (Deng et al., 2024) and WebArena (Zhou et al., 2023), have developed realistic and reproducible web environments to improve web agent capabilities. VisualWebBench (Liu et al., 2024b) has established a comprehensive evaluation framework for VLMs, focusing on UI grounding. To tackle data insufficiency issues, recent studies like SeeClick (Cheng et al., 2024) and CogAgent (Hong et al., 2023) have utilized the latest Common Crawl data to create large-scale datasets. However, these data are derived from HTML code snippets which contain plenty of noise.

This paper aims to address the aforementioned limitations of existing UI datasets by introducing an automatic LLM-based annotation pipeline. By focusing on contextual functional descriptions of elements, our pipeline aims to enhance VLM's capability of understanding users' functional intents. The advantages of our AutoGUI dataset over existing datasets are summarized in Tab. 1.

### 3 AUTOGUI: AUTOMATIC FUNCTIONALITY ANNOTATION PIPELINE

This section introduces AutoGUI, an annotation pipeline (Fig. 3) that automatically produces contextual element functionality annotations used to enhance VLMs' GUI grounding capabilities.

Table 1: **Comparing our AutoGUI dataset with existing large-scale UI datasets.** Multi-Res means the samples are collected on devices with various resolutions. Auto Anno. means the samples are collected autonomously. #Anno. means the number of annotated samples provided by the datasets.

| Dataset | UI Type | Multi Res. | Real-world Scenario | Auto Anno. | Contextual Functionality Semantics | #Anno. | Task |
|---|---|---|---|---|---|---|---|
| WebShop (Yao et al., 2022) | Web | ✗ | ✗ | ✗ | ✗ | 12k | Web Navigation |
| Mind2Web (Deng et al., 2024) | Web | ✗ | ✓ | ✗ | ✗ | 2.4k | Web Navigation |
| WebArena (Zhou et al., 2023) | Web | ✗ | ✓ | ✗ | ✗ | 812 | Web Navigation |
| S2W (Wang et al., 2021) | Mobile | ✗ | ✓ | ✗ | ✗ | 112k | Screen Summarization |
| Wid. Cap. (Li et al., 2020a) | Mobile | ✗ | ✓ | ✗ | ✗ | 163k | Element Captioning |
| PixelHelp (Li et al., 2020b) | Mobile | ✗ | ✓ | ✗ | ✗ | 187 | Element Grounding |
| RICOSCA (Li et al., 2020b) | Mobile | ✗ | ✓ | ✗ | ✗ | 295k | Action Grounding |
| MoTIF (Burns et al., 2022) | Mobile | ✗ | ✓ | ✗ | ✗ | 6k | Mobile Navigation |
| AITW (Rawles et al., 2023) | Mobile | ✗ | ✓ | ✗ | ✗ | 715k | Mobile Navigation |
| RefExp (Bai et al., 2021) | Mobile | ✗ | ✓ | ✗ | ✗ | 20.8k | Element Grounding |
| VWB (Liu et al., 2024b) | Web | ✗ | ✓ | ✗ | ✗ | 1.5k | Elem. Ground & Ref. |
| SeeClick Web (Cheng et al., 2024) | Web | ✗ | ✓ | ✓ | ✗ | 271k | Element Grounding |
| UI REC/REG (Hong et al., 2023) | Web | ✓ | ✓ | ✓ | ✗ | 400k | Box2DOM, DOM2Box |
| Ferret-UI (You et al., 2024b) | Mobile | ✓ | ✓ | ✓ | ✗ | 250k | Elem. Ground & Ref. |
| AutoGUI (ours) | Web, Mobile | ✓ | ✓ | ✓ | ✓ | 704k | Functionality Ground & Ref. |

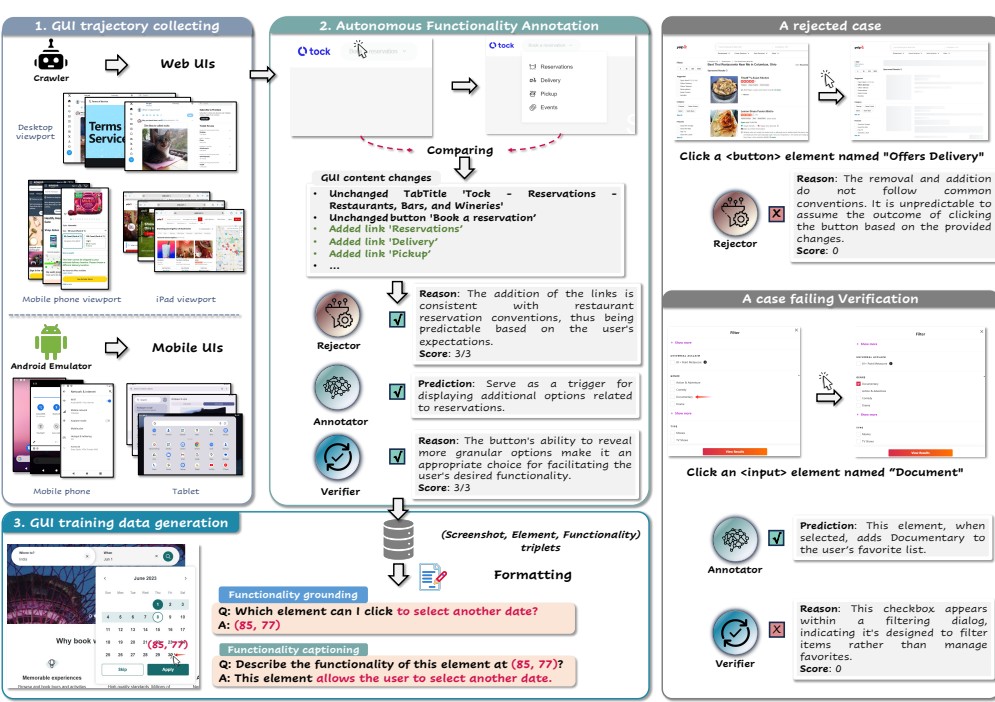

Figure 3: **The proposed pipeline for automatic UI functionality annotation.** An LLM is utilized to predict element functionality based on the UI content changes observed during the interaction. LLM-aided rejection and verification are introduced to improve data quality. Finally, the high-quality functionality annotations will be converted to instruction-following data by applying task templates.

### 3.1 COLLECTING UI INTERACTION TRAJECTORIES

Our pipeline initiates by collecting interaction trajectories, which are sequences of UI contents captured by interacting with UI elements. Each trajectory step captures all interactable elements and the accessibility tree (AXTree) that briefly outlines the UI structure, which will be used to generate functionality annotations. To amass these trajectories, we utilize the latest Common Crawl repository as the data source for web UIs and Android Emulator for mobile UIs. Note that illegal websites and Apps are excluded manually from the sources to ensure no pornographic or violent content is included in our dataset. Please refer to Sec. A.2 for collecting details and data license.

## 3.2 FUNCTIONALITY ANNOTATION BASED ON UI DYNAMICS

Subsequently, the pipeline generates functionality annotations for elements in the collected trajectories. Interacting with an element $e$, by clicking or hovering over it, triggers content changes in the UI. In turn, these changes can be used to predict the functionality $f$ of the interacted element. For instance, if clicking an element causes new buttons to appear in a column, we can predict that the element likely functions as a dropdown menu activator (an example in Fig. D). With this observation, we utilize a capable LLM (i.e., Llama-3-70B (AI@Meta, 2024)) as a surrogate for humans to summarize an element's functionality based on the UI content changes resulting from interaction. Concretely, we generate compact content differences for AXTrees before ($s_t$) and after ($s_{t+1}$) the interaction using a file-comparing library[1]. Then, we prompt the LLM to thoroughly analyze the UI content changes (addition, deletion, and unchanged lines), present a detailed Chain-of-Thoughts (Wei et al., 2022) reasoning process explaining how the element affects the UI, and finally summarize the element's functionality.

In cases where element interactions significantly transform the UI and cause lengthy differences—such as navigating to a new screen—we adjust our approach by using UI description changes instead of the AXTree differences. Specifically, we prompt the same LLM to discern the UI hierarchy, describe UI regions, and finally describe the entire UI functionality. After describing the UIs before and after the interaction, the LLM analyzes the description differences, presents reasoning, and summarizes the element's functionality. This annotation process is formulated as:

$$f = \text{LLM}(p_{\text{anno}}, s_t, s_{t+1}) \tag{1}$$

where $f$ is the predicted functionality, $p_{\text{anno}}$ is the annotation prompt (Tab. A and Tab. B). Examples of annotated elements are depicted in Fig. 4 and more annotation details are explained in Sec. A.4.

## 3.3 REMOVING INVALID SAMPLES VIA LLM-AIDED REJECTION

The collected trajectories may contain invalid samples due to broken UIs, such as incomplete UI loading. These samples are meaningless as they contain corrupted UI content and can mislead the models trained with them.

To filter out these invalid samples, we introduce an LLM-aided rejection approach. Initially, hand-written rules are used to detect obvious broken cases, such as blank UI contents, UIs containing elements indicating content loading, and interaction targets outside of UIs. While these obvious cases constitute a large portion of the invalid samples, there are a few types that are difficult to detect with hand-written rules. For instance, interacting with a "view more" button might unexpectedly redirect the user to a login page instead of the desired information page due to website login restrictions. To identify these challenging samples, we prompt the annotating LLM to also act as a rejector. Specifically, the LLM takes the UI content changes, generated using a file-comparing library, as input, provides detailed reasoning on whether the changes are meaningful for predicting the element's functionality, and finally outputs predictability scores ranging from 0 to 3. This process is formulated as follows:

$$score = \text{LLM}(p_{\text{reject}}, e, s_t, s_{t+1}) \tag{2}$$

where $p_{\text{reject}}$ is the rejection prompt (Tab. C).

This approach ensures that clear and predictable samples receive higher scores, while those that are ambiguous or unpredictable receive lower scores. For instance, if a button labeled "Show More", upon interaction, clearly adds new content, this sample will considered to provide sufficient changes that can anticipate the content expansion functionality and will get a score of 3. Conversely, if clicking on a "View Profile" link fails to display the profile possibly due to web browser issues, this unpredictable sample will get a score less than 3.

After implementing empirical experiments, we deploy this LLM-based rejector to discard the bottom 30% of samples based on their scores to strike a balance between the elimination of invalid samples and the preservation of valid ones (More details in Sec. A.6). The samples that pass the hand-written rules and the LLM rejector are subsequently submitted for functionality annotation. Please see representative rejection examples in Fig. H.

---

[1]https://docs.python.org/3/library/difflib.html

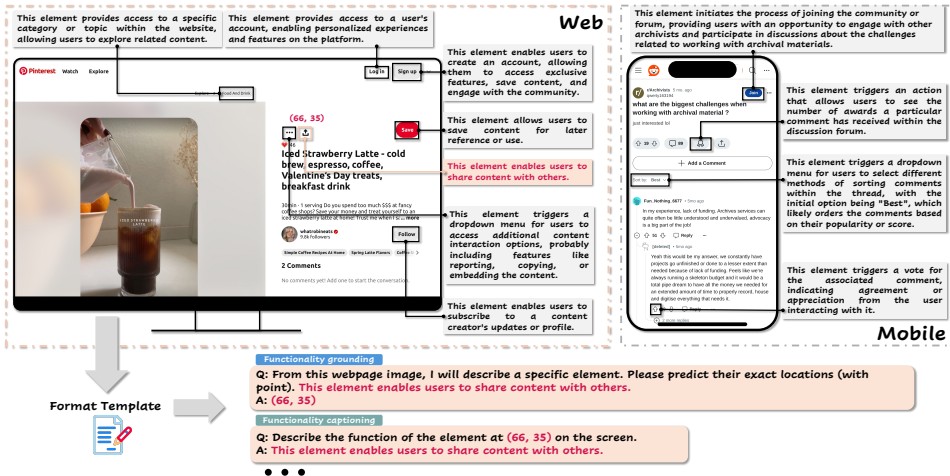

Figure 4: Element functionality annotations generated by the proposed AutoGUI pipeline for both web and mobile viewpoints.

## 3.4 IMPROVING ANNOTATION QUALITY VIA LLM-BASED VERIFICATION

The functionality annotations produced by the LLM probably contain incorrect, ambiguous, and hallucinated samples (See a case in Fig. 3), which probably misleads the trained VLMs and compromises evaluation accuracy. To improve dataset quality, we prompt LLMs to verify the annotations by checking whether the targeted element $e$ fulfills the intent of the annotated functionality $f$. This process presents the LLMs with the interacted element, its UI context, the UI changes induced by this element, and the functionality generated in the previous annotation process. The LLMs are then tasked with analyzing the UI content changes before predicting whether the interacted element aligns with the given functionality. If the LLMs determine that the interacted element fulfills the functionality given its UI context, the LLMs will grant a full score (An example in Fig. I). If the interacted element is considered to mismatch the functionality, this functionality can be seen as incorrect as this mismatch indicates that it may not accurately reflect the element's actual role within the UI context.

To mitigate the potential biases in LLMs (Panickssery et al., 2024; Zheng et al., 2023; Bai et al., 2024), two different LLMs (i.e., Llama-3-70B (AI@Meta, 2024) and Mistral-7B-Instruct-v0.2 (Jiang et al., 2023)) are employed as verifiers and prompted to output 0-3 scores. The scoring process is formulated as follows:

$$score = \text{LLM}(p_{\text{verify}}, e, f, s_t, s_{t+1}) \tag{3}$$

where $p_{\text{verify}}$ denotes the verification prompt (Tab. D). Only if the two scores are both 3s do we consider the functionality label correct (More details in Sec. A.7). Although this filtering approach seems stringent, we can make up the number of annotations through scaling.

## 3.5 FUNCTIONALITY GROUNDING AND REFERRING TASK GENERATION

After rejecting, annotating, and verifying, we obtain a high-quality UI functionality dataset containing triplets of {UI screenshot, Interacted element, Functionality}. To convert this dataset into an instruction-following dataset for training and evaluation, we generate functionality grounding and referring tasks using diverse prompt templates (see Tab. E). To mitigate the difficulty of predicting absolute values for various resolutions, the coordinates of element bounding boxes are all normalized within the range $[0, 99]$ (see Fig. 4 for examples).

## 3.6 EXPLORE THE AUTOGUI DATASET

The AutoGUI pipeline finally collects 22.4k trajectories, from which we select 2k grounding samples (evenly divided between web and smartphone views) as the test set and remove the trajectories to which these samples belong. Subsequently, 702k samples are randomly selected from the remaining instances to constitute the training set. The statistics of our dataset in Tab. 2 and Sec. A.1 show that our dataset covers diverse UIs and exhibits variety in lengths and functional semantics of the

Table 2: **The statistics of the AutoGUI datasets.** The Anno. Tokens and Avg. Words columns show the total number of tokens and the average number of words for the functionality annotations regardless of task templates. The Domains/Apps column shows the number of unique web domains/mobile Apps involved in each split.

| Split | #Tasks | Anno. Tokens | Avg. Words | Domains/Apps | Device Ratio |
|-------|--------|--------------|------------|--------------|--------------|
| Train | 702k | 17.9M | 23.1 | 916 | Web: 54.6%, Mobile: 45.4% |
| Test | 2k | 53.4k | 22.5 | 299 | Web: 50%, Mobile: 50% |

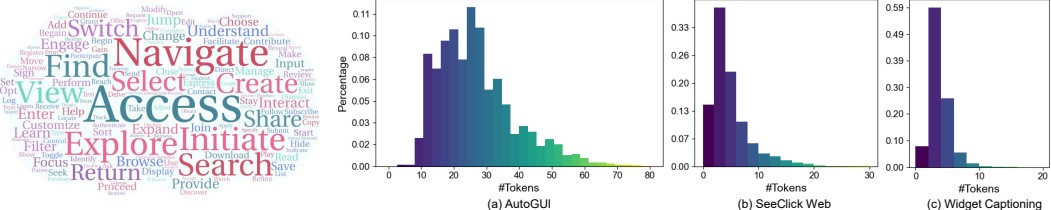

Figure 5: **Diversity of the AutoGUI dataset. Left**: The word cloud illustrates the ratios of the verbs representing the main intents in the functionality annotations. **Right**: Comparing the distributions of the annotation token numbers for our AutoGUI training split, SeeClick Web training data (Cheng et al., 2024), and Widget Captioning (Li et al., 2020a). The comparison demonstrates that our dataset covers significantly more diverse task lengths.

annotations. Moreover, our dataset presents a unique ensemble of research challenges for developing generalist web agents in real-world settings. As shown in Tab. 1 and Fig. 2, our dataset distinguishes itself from existing literature by providing functionality-rich data as well as tasks that require VLMs to discern the contextual functionalities of elements to achieve high grounding accuracy.

# 4 ANALYSIS OF DATA QUALITY

This section analyzes the reliability of the proposed annotation pipeline and data quality.

**Comparison with Human Annotation** To demonstrate the superiority of the proposed automatic annotation pipeline based on open-source LLMs, $N = 145$ samples (99 valid and 46 invalid) are randomly selected as a testbed for comparing the annotation correctness of a trained human annotator and the pipeline. Here, correctness is defined as $Correctness = C/(N - R)$, where $C$ and $R$ denote the numbers of correctly annotated and rejected samples, respectively. The denominator subtracts the number of rejected samples as we are more interested in the percentage of correct samples after rejecting the samples considered invalid by the annotator. The authors thoroughly check the annotation results according to the three criteria in Fig. 6: 1. Context-specificity. The functionality annotations must include context-specific descriptions to ensure one-to-one mapping between the element and its annotation. 2. Appropriate details. Avoid detailing unnecessary aspects of the UIs to keep the description focused on functionality. 3. No hallucination. The annotations must not include information not grounded in the visual context of the UIs. See more details in Sec. B.1.

After experimenting with three runs, Tab. 3 shows that the proposed AutoGUI pipeline achieves high correctness comparable to the trained human annotator (r6 vs. r1). Without rejection and verification (r2), AutoGUI is inferior as it cannot recognize invalid samples. Notably, simply using the rules written by the authors can improve the correctness, which is further enhanced with the LLM-aided rejector (r4 vs. r3). Moreover, utilizing the annotating LLM itself to self-verify its annotations helps AutoGUI surpass the trained annotator (r5 vs. r1). Introducing another LLM verifier (i.e., Mistral-7B-Instruct-v0.2) brings a slight increase which results from Mistral recognizing Llama-3-70B's incorrect descriptions of how dropdown menu options work. Overall, these results justify the efficacy of the AutoGUI annotation pipeline.

Qualitatively comparing the annotation patterns of the human and AutoGUI (Fig. O), we find that AutoGUI employs the strong LLM to generate more detailed and clear annotations which would take

- Criterium 1: Context-specificity
  Context-related description
  ☑ This element opens a date-picker to **select a check-in date**.
  ☒ This element reveals a calendar.

- Criterium 2: Appropriate details
  Unnecessary details
  ☑ This element triggers a date picker for users to select their check-in date
  ☒ This element **switches the focus from the 'Destination' input field to the 'Check-in Date' field**, providing a date picker.

- Criterium 3: No hallucination
  Hallucinated functionality
  ☑ This element checks an item shown in a filter dialog.
  ☒ This element **immediately filters and reorganizes** the content of the UI.

Figure 6: The checking criteria used for comparing AutoGUI pipeline and the human annotator.

Table 3: **Comparing the AutoGUI and human annotator.** AutoGUI with the proposed rejection and verification achieves annotation correctness comparable to trained human annotators. One LLM means Llama-3-70B and Two LLMs include Mistral-7B-Instruct-v0.2 as well.

| No. | Annotator | Rejector | Verifier | Correctness |
|-----|-----------|----------|----------|-------------|
| r1 | Human | - | - | 95.5% |
| r2 | Llama-3-70B | - | - | 64.5% |
| r3 | Llama-3-70B | Rules | - | 83.1% |
| r4 | Llama-3-70B | Rules+LLM | - | 94.4% |
| r5 | Llama-3-70B | Rules+LLM | One LLM | 96.0% |
| r6 | Llama-3-70B | Rules+LLM | Two LLMs | **96.7%** |

significantly more time for the human annotator. This result suggests that the AutoGUI pipeline can lessen the burden of collecting data for training UI-VLMs.

**Impact of LLM Output Uncertainty** The uncertainty of LLM outputs manifests in annotation, rejection, and verification, possibly impacting the quality of the AutoGUI dataset. To evaluate this impact, we first sample 100 valid samples to test the AutoGUI pipeline for three runs. The consistency rate is 94.5%, indicating that 94.5% of the samples possess consistent annotation outcomes (i.e. correct or incorrect) across the runs. We also test the LLM-aided rejector with 46 invalid samples and find that the rejection consistency over three runs is 79.3%. This indicates that LLM uncertainty impacts this rejection process. Nevertheless, this impact is minor due to the low prevalence of invalid samples (4% of all samples) that fail the hand-written rules.

In summary, AutoGUI exhibits annotation correctness comparable to that of human annotators and LLM output uncertainty poses a minor impact on the AutoGUI annotation process.

## 5 FINE-TUNING EXPERIMENTS

This section validates that our dataset can enhance the GUI grounding capabilities of VLMs and that the proposed functionality grounding and referring are effective fine-tuning tasks.

### 5.1 EXPERIMENTAL SETTINGS

**Evaluation Benchmarks** We base our evaluation on the UI grounding benchmarks for various scenarios: **FuncPred** is the test split from our collected functionality dataset. This benchmark requires a model to locate the element specified by its functionality description. **ScreenSpot** (Cheng et al., 2024) is a benchmark comprising test samples on mobile, desktop, and web platforms. It requires the model to locate elements based on short instructions. **RefExp** (Bai et al., 2021) is to locate elements given crowd-sourced referring expressions. **VisualWebBench (VWB)** (Liu et al., 2024b) is a comprehensive multi-modal benchmark assessing the understanding capabilities of VLMs in web scenarios. We select the element and action grounding tasks from this benchmark. To better align with high-level semantic instructions for potential agent requirements and avoid redundancy evaluation with ScreenSpot, we use ChatGPT to expand the OCR text descriptions in the original task instructions, such as *Abu Garcia College Fishing* into functionality descriptions like *This element is used to register for the Abu Garcia College Fishing event*. **MOTIF** (Burns et al., 2022) requires an agent to complete a natural language command in mobile Apps. For all of these benchmarks, we report the grounding accuracy (%): $\text{Acc} = \sum_{i=1}^{N} \mathbf{1}\left(\text{pred}_i \text{ inside GT bbox}_i\right)/N \times 100$ where $\mathbf{1}$ is

Table 4: **Element grounding accuracy on the used benchmarks.** We compare the base models fine-tuned with our AutoGUI data and representative open-source VLMs. The results show that the two base models (i.e. Qwen-VL and SliME-8B) obtain significant performance gains over the benchmarks after being fine-tuned with AutoGUI data. Moreover, increasing the AutoGUI data size consistently improves grounding accuracy, demonstrating notable scaling effects. † means the metric value is borrowed from the benchmark paper. ∗ means using additional SeeClick training data.

| Type | Model | Size | FuncPred | VWB EG | VWB AG | MoTIF | RefExp | ScreenSpot |
|---|---|---|---|---|---|---|---|---|
| General | LLaVA-1.5 (Liu et al., 2023b) | 7B | 3.2 | 12.1† | 13.6† | 7.2 | 4.2 | 5.0 |
| | LLaVA-1.5 (Liu et al., 2023b) | 13B | 5.8 | 16.7 | 9.7 | 12.3 | 20.3 | 11.2 |
| | LLaVA-1.6 (Liu et al., 2024a) | 34B | 4.4 | 19.9 | 17.0 | 7.0 | 29.1 | 10.3 |
| | SliME (Zhang et al., 2024b) | 8B | 3.2 | 6.1 | 4.9 | 7.0 | 8.3 | 13.0 |
| | Qwen-VL (Bai et al., 2023) | 10B | 3.0 | 1.7 | 3.9 | 7.8 | 8.0 | 5.2† |
| UI-VLM | Qwen2-VL (Bai et al., 2023) | 7B | 7.8 | 3.9 | 3.9 | 16.7 | 32.4 | 26.1 |
| | CogAgent (Hong et al., 2023) | 18B | 29.3 | 55.7 | 59.2 | 24.7 | 35.0 | 47.4† |
| | SeeClick (Cheng et al., 2024) | 10B | 19.8 | 39.2 | 27.2 | 11.1 | 58.1 | 53.4† |
| Finetuned | Qwen-VL-AutoGUI25k | 10B | 14.2 | 12.8 | 12.6 | 10.8 | 12.0 | 19.0 |
| | Qwen-VL-AutoGUI125k | 10B | 25.5 | 23.2 | 29.1 | 11.5 | 14.9 | 32.0 |
| | Qwen-VL-AutoGUI702k | 10B | 43.1 | 38.0 | 32.0 | 15.5 | 23.9 | 38.4 |
| | Qwen-VL-AutoGUI702k∗ | 10B | 50.0 | 56.2 | 45.6 | 21.0 | 51.5 | 54.2 |
| Finetuned | SliME-AutoGUI25k | 8B | 28.0 | 14.0 | 10.6 | 14.3 | 18.4 | 27.2 |
| | SliME-AutoGUI125k | 8B | 39.9 | 22.0 | 12.0 | 17.8 | 22.1 | 35.0 |
| | SliME-AutoGUI702k | 8B | 62.6 | 25.4 | 13.6 | 20.6 | 26.7 | 44.0 |

an indicator function and $N$ is the number of test samples. This formula denotes the percentage of samples with the predicted points lying within the bounding boxes of the target elements.

**Training Details** We select Qwen-VL-10B (Bai et al., 2023) and SliME-8B (Zhang et al., 2024b) as the base models and fine-tune them on 25k, 125k, and 702k samples of the AutoGUI training data to investigate how the AutoGUI data enhances the UI grounding capabilities of the VLMs. The models are fine-tuned on 8 A100 GPUs for one epoch. We follow SeeClick (Cheng et al., 2024) to fine-tune Qwen-VL with LoRA (Hu et al., 2022) and follow the recipe of SliME (Zhang et al., 2024b) to fine-tune it with only the visual encoder frozen (More details in Sec. B.2).

**Compared VLMs** We compare with both general-purpose VLMs (i.e., LLaVA series (Liu et al., 2023b; 2024a), SliME (Zhang et al., 2024b), and Qwen-VL (Bai et al., 2023)) and UI-oriented ones (i.e., Qwen2-VL (Wang et al., 2024a), SeeClick (Cheng et al., 2024), CogAgent (Hong et al., 2023)). SeeClick finetunes Qwen-VL with around 1 million data combining various data sources, including a large proportion of human-annotated UI grounding/referring samples. CogAgent is trained with a huge amount of text recognition, visual grounding, UI understanding, and publicly available text-image datasets, such as LAION-2B (Schuhmann et al., 2022). During the evaluation, we manually craft grounding prompts suitable for these VLMs.

## 5.2 EXPERIMENTAL RESULTS AND ANALYSIS

**A) AutoGUI functionality annotations effectively enhance VLMs' UI grounding capabilities and achieve scaling effects.** We endeavor to show that the element functionality data autonomously collected by AutoGUI contributes to high grounding accuracy. The results in Tab. 4 demonstrate that on all benchmarks the two base models achieve progressively rising grounding accuracy as the functionality data size scales from 25k to 702k, with SliME-8B's accuracy increasing from merely **3.2** and **13.0** to **62.6** and **44.0** on FuncPred and ScreenSpot, respectively. This increase is visualized in Fig. K showing that increasing AutoGUI data amount leads to more precise localization performance.

After fine-tuning with AutoGUI 702k data, the two base models surpass SeeClick, the strong UI-oriented VLM on FuncPred and MOTIF. We notice that the base models lag behind SeeClick and CogAgent on ScreenSpot and RefExp, as the two benchmarks contain test samples whose UIs cannot be easily recorded (e.g., Apple devices and Desktop software) as training data, causing a domain gap. Nevertheless, SliME-8B still exhibits noticeable performance improvements on ScreenSpot and RefExp when scaling up the AutoGUI data, suggesting that the AutoGUI data helps to enhance grounding accuracy on the out-of-domain tasks.

Table 5: **Comparing the AutoGUI functionality annotation type with existing types**. Qwen-VL is fine-tuned with the three annotation types. The results show that our functionality data leads to superior grounding accuracy compared with the naive element-HTML data and the condensed functionality annotations.

| Data Size | Variant | FuncPred | RefExp | ScreenSpot |
|---|---|---|---|---|
| 25k | w/ Elem-HTML data | 5.3 | 4.5 | 5.7 |
| | w/ Condensed Func. Anno. | 3.8 | 3.0 | 4.8 |
| | w/ Func. Anno. (Ours full) | **21.1** | **10.0** | **16.4** |
| 125k | w/ Elem-HTML data | 15.5 | 7.8 | 17.0 |
| | w/ Condensed Func. Anno. | 14.1 | 11.7 | 23.8 |
| | w/ Func. Anno. (Ours full) | **24.6** | **12.7** | **27.0** |

To further unleash the potential of the AutoGUI data, the base model, Qwen-VL, is finetuned with the combination of the AutoGUI and SeeClick UI-grounding data. This model becomes the new state-of-the-art on FuncPred, ScreenSpot, and VWB EG, surpassing SeeClick and CogAgent. This result suggests that our AutoGUI data can be mixed with existing UI grounding training data to foster better UI grounding capabilities.

In summary, our functionality data can endow a general VLM with stronger UI grounding ability and exhibit clear scaling effects as the data size increases.

**B) Our functionality annotations are effective for enhancing UI grounding capabilities.** To assess the effectiveness of functionality annotations, we compare this annotation type with two existing types: 1) **Naive element-HTML pairs**, which are directly obtained from the UI source code (Hong et al., 2023) and associate HTML code with elements in specified areas of a screenshot. Examples are shown in Fig. 2. To create these pairs, we replace the functionality annotations with the corresponding HTML code snippets recorded during trajectory collection. 2) **Brief functionality descriptions** that are generated by prompting GPT-4o-mini[2] to condense the AutoGUI functionality annotations. For example, a full description such as *'This element provides access to a documentation category, allowing users to explore relevant information and guides'* is shortened to *'Documentation category access'*.

After experimenting with Qwen-VL (Bai et al., 2023) at the 25k and 125k scales, the results in Tab. 5 show that fine-tuning with the complete functionality annotations is superior to the other two types. Notably, our functionality annotation type yields the largest gain on the challenging FuncPred benchmark that emphasizes contextual functionality grounding. In contrast, the Elem-HTML type performs poorly due to the noise inherent in HTML code (e.g., numerous redundant tags), which reduces fine-tuning efficiency. The condensed functionality annotations are inferior, as the consensing loses details necessary for fine-grained UI understanding. In summary, the AutoGUI functionality annotations provide a clear advantage in enhancing UI grounding capabilities.

## 5.3 Failure Case Analysis

After analyzing the grounding failure cases, we identified several failure patterns in the fine-tuned models: a) difficulty in accurately locating small elements; b) challenges in distinguishing between similar but incorrect elements; and c) issues with recognizing icons that have uncommon shapes. Please refer to Sec. C.2 for details.

## 6 Conclusion

We propose AutoGUI, a scalable and automatic annotation pipeline aimed to produce massive UI element functionality annotations used to enhance UI understanding capabilities of open-source VLMs. The pipeline prompts an open-source LLM to generate element functionalities based on the UI content changes induced by interacting with the elements. To guarantee high quality, LLM-aided rejection and verification are introduced to remove invalid samples. Fine-tuned with the data collected by AutoGUI, the base models obtain strong UI grounding ability and exhibit data scaling effects. We hope that AutoGUI will open up possibilities for advancing the field of general UI agents.

---

[2]https://openai.com/index/gpt-4o-mini-advancing-cost-efficient-intelligence/

REPRODUCIBILITY STATEMENT

The AutoGUI annotation pipeline is fully reproducible. The prompts used for annotating, LLM-aided rejection, and verification are listed in Tab. A, Tab. C, and Tab.D, respectively. The fine-tuning experiments are also reproducible, as we employ the training code repositories of open-source VLMs, i.e., SeeClick and SliME. Readers can download our data and use these training code repos to reproduce our models.

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

# A APPENDIX

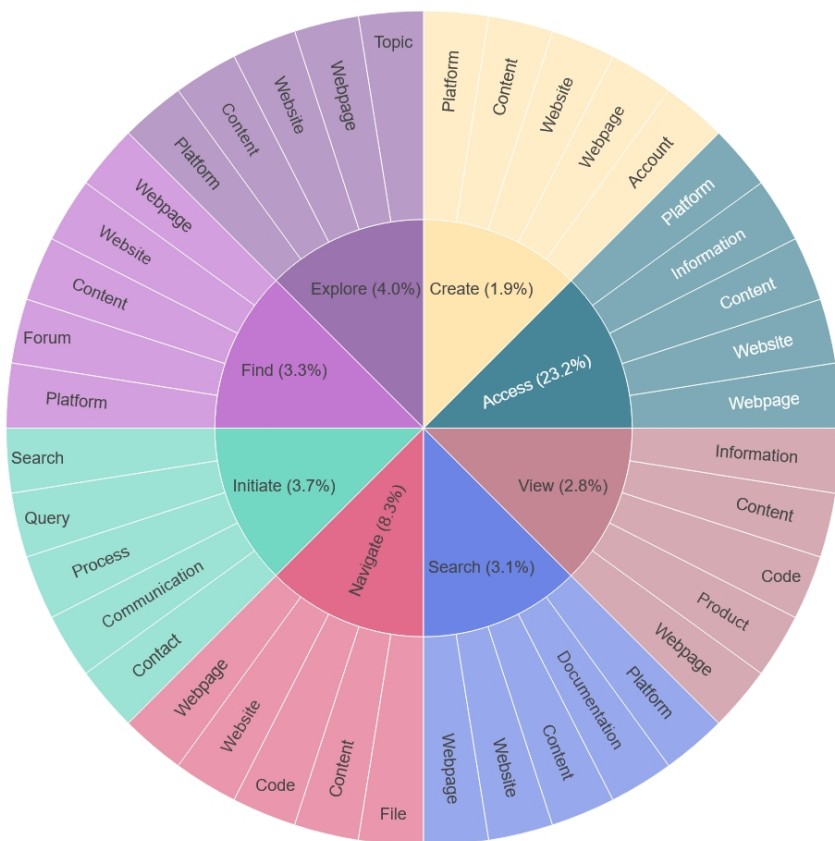

Figure A: **Diversity of the verb-noun phrases of the AutoGUI dataset.** The top 10 verbs and their top 5 following nouns are displayed. This diagram shows that our dataset contains diverse tasks that involve various UI functions.

The appendix comprises the following sections:

Section A: Details for implementation details for the autonomous annotation pipeline, including dataset statistics, visualized annotation pipeline, and LLM prompts.

Section B: Details for model implementation and training.

Section C: Additional experimental analysis including analysis of successful and failure cases on two benchmarks.

Section D and E: Limitations and Potential Societal Impact.

## A    DETAILS OF THE AUTOGUI PIPELINE

### A.1    EXTRA STATISTICS OF THE AUTOGUI DATASET

Fig. A visualizes the verb-noun statistics of the AutoGUI dataset, highlighting its extensive coverage of diverse UI functionalities. Fig. B lists the top 50 most frequent top-level domains in the AutoGUI dataset, showing that the AutoGUI dataset involves a broad spectrum of real-world scenarios, including technology (e.g., apple.com), entertainment (e.g., tiktok.com), office (e.g., outlook.com), news (e.g., medium.org), and finance (e.g., paypal.com).

### A.2    RECORDING INTERACTION TRAJECTORIES ON WEB

**Interactive Crawler for Common Crawl** We design an in-house web crawler that interacts with most elements rendered on the web page. In contrast with existing methods which contain information

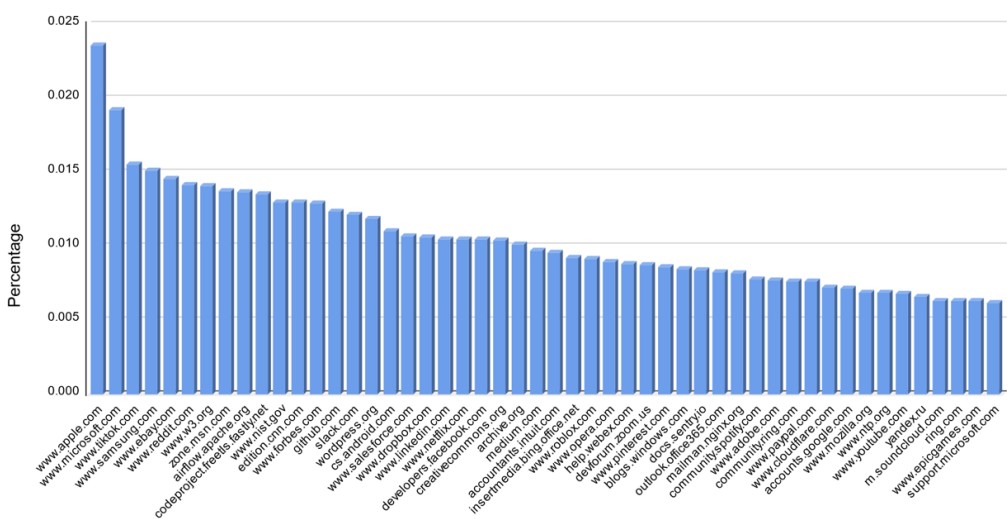

Figure B: The top-50 most frequent top-level domains in the AutoGUI dataset.

for elements on the initial static web page for a given URL, our crawler randomly interacts with a rendered web page for **multiple steps** within a given action horizon $T_{act}$ to collect UI data with abundant functional semantics. Fig. C compares the proposed AutoGUI and the existing annotation methods. We empirically set $T_{act} = 10$ in all our recordings. Therefore, our interactive crawler could collect functionality of elements that are not visible to static pages, including nested drop-down menus, date and location selectors, and secondary menus.

**Data Source and Data Format** To incorporate a wide basis of web pages, we first obtain a list of the top-200 most visited domains [3] and manually remove content delivery network (CDN) and not safe for work (NSFW) sites. We use URLs in this curated list as seeds to query the Common Crawl index [4] to find additional URLs with maximum sub-domain and path diversity. Querying URLs from the Common Crawl index ensures that our crawler respects each site's robots.txt file, making the dataset collection process legally safe. By obeying the directives in robots.txt, we avoid potential legal issues associated with unauthorized web scraping. For each web page, we collect the following data:

- Screenshot image of the rendered page

- Accessible Tree (AXTree) text representing the page's accessibility structure

- HTML source code of the page

- Accessible Node (AXNode) text for the specific element our crawler interacted with at each step

### A.3 RECORDING INTERACTION TRAJECTORIES ON ANDROID DEVICES

We also implement an in-house crawler that interacts with multiple emulated Google Pixel phones. The phones are reset to different starting UIs before a script randomly interacts with these phones to record trajectories. To improve data diversity, the starting UIs include the home page, drop-down panel, settings page, and Apps drawer.

Similar to webpage HTML, mobile phone UIs are rendered with XML code, which is cleaned and converted to AXTree-like content before being used to annotate functionalities.

---

[3] https://tranco-list.eu/
[4] https://index.commoncrawl.org/

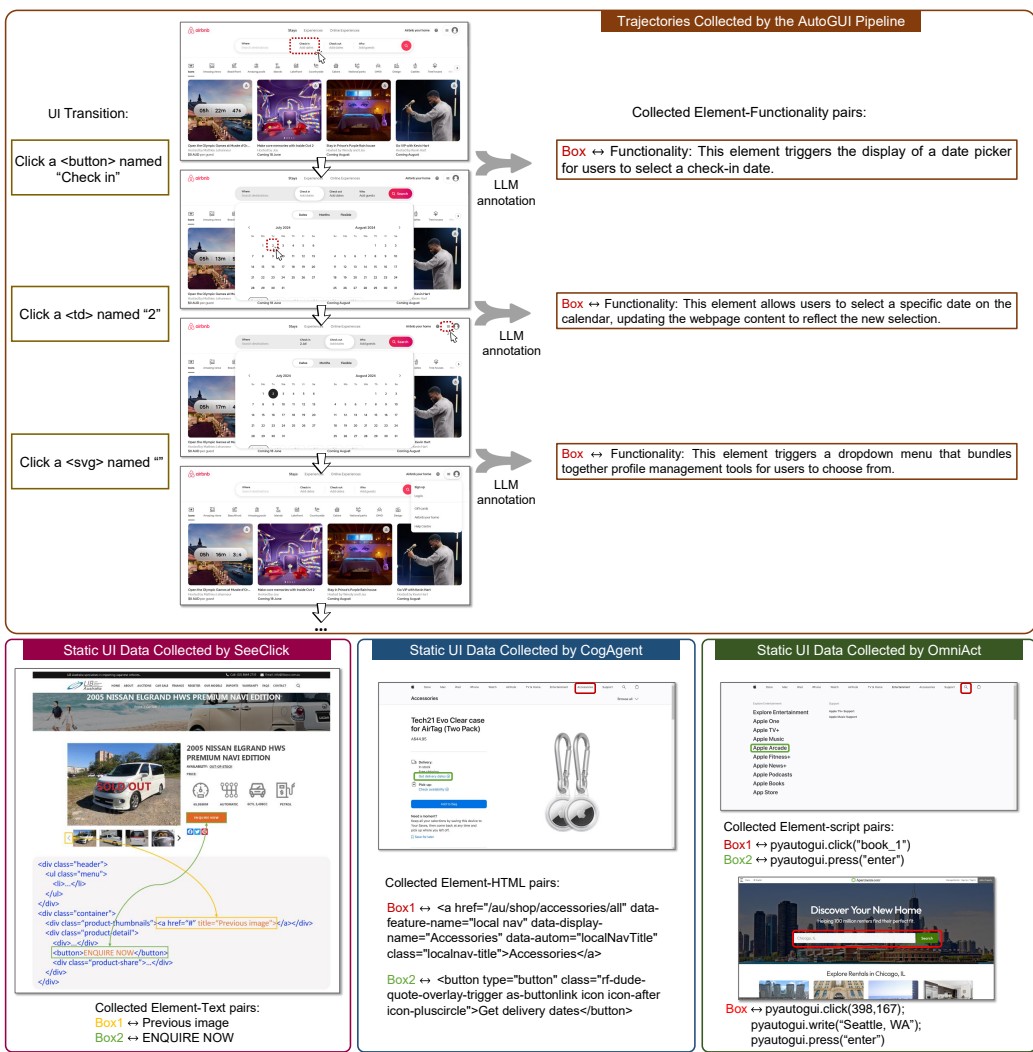

Figure C: **Comparing the proposed AutoGUI annotation pipeline with existing methods.** AutoGUI is able to manipulate real UIs and interact with elements hidden beneath deeper levels (e.g., the buttons hidden in collapsed dropdown menus), thereby collecting considerably rich element-functionality annotations from the immense UI resources on the Internet. In contrast, SeeClick Cheng et al. (2024) only uses static webpages and collects static element-text pairs. Likewise, CogAgent collects static element-HTML pairs while OmniAct generates Python scripts only for visible elements. These three existing methods can only annotate visible static UI elements and ignore the rich UI functional semantics entailed in interaction trajectories which are provided by our AutoGUI pipeline in abundance.

## A.4 FUNCTIONALITY ANNOTATION DETAILS

The AutoGUI pipeline utilizes UI content changes to predict the functionalities of the interacted elements. For interactions that manipulate the existing UI, the pipeline analyzes differences in the AXTrees to annotate functionalities. Conversely, when interactions result in navigation to a new UI, the pipeline examines changes in UI descriptions to guide the annotation process. Details on these methodologies are outlined below:

**UI manipulation case** We use a file-comparison library, DiffLib, to generate line-by-line differences of the AXtrees before and after interactions. To balance efficiency with annotation integrity, we limit the differences to 250 lines. In addition to the standard markings by DiffLib—addition, deletion, and unchanged status—we incorporate two additional change markers: 'Repositioning' and

Table A: The functionality annotation prompt used in the AutoGUI pipeline in UI manipulation cases.

---

(Requirements for annotation)

Objective: As an Internet expert, your task is to describe the usage and functionality of a webpage element based on the changes observed in the webpage contents before and after interacting with the element.

Instructions:

1. You will be shown line-by-line differences between the webpage content before and after interacting with the element. Here's what each prefix indicates:

Unchanged: Lines that are identical before and after the interaction.

Added: New lines that appear after the interaction.

Deleted: Lines that were present before the interaction but removed afterward.

Renaming: Lines indicating elements that were renamed due to the interaction.

Attribute Update: Lines showing elements whose attributes were updated during the interaction.

Repositioned: Elements that were moved to a different part of the webpage.

2. You MUST thoroughly analyze the changes in webpage content (Added, Deleted, Unchanged lines) caused by interacting with the element, present a detailed reasoning process elucidating how the element affects the webpage, and finally summarize the element's overall purpose based on your analysis

3. Avoid detailing every functionality of the webpage element. Instead, focus on describing its broader impact on the webpage experience. For example, if interacting with a "Products" button reveals a dropdown menu, do not catalog the subsequent webpage changes in exhaustive detail.

4. Your output MUST follow this format:

Reasoning: (Examine the webpage variation carefully to figure out how the interacted element changes the webpage)

Summary: This element ... (Provide a concise high-level description of the element's function. This description should contain the meaningful feature of the element in its context.)

5. Avoid mentioning specific elements from the webpage before interaction in the Summary. Instead, focus directly on the outcome of the interaction. For instance, rather than detailing a transition from one element to another, like "This element switches the focus from the 'Destination' input field to the 'Check-in Date' field, providing a date picker," simplify it to "This element triggers a date picker for users to select their check-in date." This approach keeps the description concise and focused on functionality, eliminating unnecessary details.

6. For simplicity, you MUST not mention the text of the element in the summary.

Now it's your turn.

Element interaction: After clicking a <link> element named "Community submenu" (an example of the interaction target), the webpage exhibits the following variations:

(UI content changes)

Unchanged button 'Extend submenu' expanded: False

Unchanged StaticText 'Extend'

Unchanged button 'Learn submenu' expanded: False

Unchanged StaticText 'Learn'

Before Attribute Update button 'Community submenu' expanded: False

After Attribute Update button 'Community submenu' focused: True expanded: True

Unchanged StaticText 'Community'

Added link 'Make WordPress'

Added link 'Photo Directory'

Added link 'Five for the Future'

Added link 'Events'

Added link 'Job Board'

Unchanged button 'About submenu' expanded: False

Unchanged StaticText 'About'

Unchanged button 'Open Search' hasPopup: dialog

Unchanged link 'Get WordPress'

Now please think step-by-step, examine the content variation in detail, and finally summarize the high-level usage of the interacted element. Do follow the output format "Reasoning: ... Summary: This element ...".

---

(LLM response)

Reasoning: After interacting with the "Community submenu" button, the webpage content changes significantly. The button's "expanded" attribute is updated to "True", indicating that the button is now expanded. Additionally, the button gains focus, as indicated by the "focused: True" attribute. The most notable change, however, is the appearance of new. These new elements suggest that the button has triggered the display of a submenu or dropdown list related to the "Community" theme.

Summary: This element reveals a submenu of community-related links and resources.

---

'Attribute Update'. These markers provide detailed information about UI content changes, essential for representing realistic structural variations. For example, without the attribute update marker, a clicked menu icon would erroneously appear as both deleted and added in the difference output, despite the menu element remaining in place. An example of this case is shown in Fig. D. The used prompt is shown in Tab. A.

**UI navigation case** When an interacted element causes navigation to a new UI, the resultant changes are often extensive, potentially exceeding the context limit of an LLM and complicating the analysis of these changes. To handle this situation, UI description changes are used to predict functionalities. Concretely, the LLM is initially prompted to describe the UIs before and after interaction given UI AXTrees as input. Subsequently, the LLM then uses these descriptions to analyze content changes and predict the functionality of the interacted element. The description length of the AXTree is limited to 150 lines. An illustration of this process is shown in Fig. E. The corresponding prompt is detailed in Tab. B.

A.5 DETAILS OF REJECTING INVALID SAMPLES VIA HAND-WRITTEN RULES

To clarify the hand-written rules used in the process of removing invalid samples: (1) **Removing blank GUIs**. We remove blank GUIs by verifying whether the accessibility tree contains more than

Table B: The functionality annotation prompt used in the AutoGUI pipeline in UI navigation cases. This example shows how the LLM

---

(Requirements for annotation)

Objective: Your mission, as a digital navigation specialist, is to deduce and articulate the function and usage of a specific webpage element. This deduction should be based on your analysis of the differences in webpage content before and after interacting with said element.

Instructions:

1. You will be given descriptions of a webpage before and after interaction with an element. Your primary task is to meticulously analyze the differences in content resulting from this interaction to understand what the functionality of the element is in the webpage context.

2. You must present a detailed reasoning process before finally summarizing the element's overall purpose based on your analysis.

3. Prioritize examining changes in the webpage's regional content over individual element variations. This approach will provide a more holistic view of the element's impact on the webpage.

4. You should emphasize on the main content changes and pay less attention to less meaningful regions, such as headers, navigation bars, and footers.

5. Your output MUST follow this format:

Reasoning: (Examine the webpage variation carefully to figure out how the interacted element changes the webpage)

Summary: This element ... (Provide a high-level description of the element's functionality. This description should contain the meaningful feature of the element in its context.)

6. Avoid mentioning specific elements from the webpage prior to interaction in the Summary. Instead, focus directly on the outcome of the interaction. For instance, rather than detailing element changes, like "This element triggers the disappearance of the header and language selector elements and the emergence of a login form", simplify it to "This element triggers the display of a login page." This approach keeps the summary concise and focused on functionality, eliminating unnecessary details.

7. The final summary must emphasize the distinctive functionality of the element and not be too vague. For example, "This element redirects the user to a specific website, allowing them to access its content and services" is too broad and lacks meaningful detail. Instead, it should precisely describe how the element provides users with unique value.

Now it's your turn.

The description of the initial webpage:

(An example description of the UI before the interaction)

Region 1 (Header): Navigation and login functionality

Region 2 (Main Content): Announcing updates to Instagram Direct Message, providing a blog post with a title, date, and content

Region 3 (Footer): Additional links and buttons (less meaningful region, summarized in a single sentence)

Overall Functionality: A blog post webpage announcing updates to Instagram Direct Message, providing a title, date, and content, with navigation and login functionality at the top.

After clicking a <link> element named "" (an example of the interaction target), we jump to a new webpage.

The description of the new webpage is:

(An example description of the UI after the interaction)

Region 1 (Header): Displays the webpage title and heading.

Region 2 (Login Form): Allows users to log in to their Facebook account by providing their email address or phone number and password.

Region 3 (Forgot Password): Provides a link to recover a forgotten account password.

Region 4 (Create New Account): Offers an option to create a new Facebook account.

Overall Functionality: Enables users to access their existing Facebook account or create a new one, providing a gateway to the social media platform.

Now please think step-by-step, examine the content variation in detail, and finally summarize the high-level usage of the interacted element. Do follow the output format "Reasoning: ... Summary: This element ...".

---

(LLM response)

Reasoning: Upon examining the webpage variations, I notice that the initial webpage is a blog post announcing updates to Instagram Direct Message, with navigation and login functionality at the top. After clicking the '<link>' element named "", the webpage transitions to a new page with a Facebook title and heading. The main content of the new webpage is a login form, allowing users to log in to their Facebook account. Additionally, there are options to recover a forgotten password and create a new Facebook account.

Summary: This element redirects the user to a Facebook login page, allowing them to access their existing account or create a new one.

one node. If no nodes are present, the GUI is considered blank. (2) **Removing UIs containing elements indicating content loading**. GUIs containing elements indicative of content loading (e.g., keywords such as "loading", "please wait", or "refreshing") are excluded. These keywords typically suggest that the content has not fully loaded and may affect the validity of the sample. (3) **Removing interaction targets outside of screens**. Occasionally, part of the UI, including the interacted element, may fail to be captured. We filter out GUIs where interaction targets appear outside of the visible screen area. This is determined by checking whether the interacted element exists within the bounds of the recorded accessibility tree. Note that these rules are designed mainly for the domains from which we collected GUI metadata. Nevertheless, one can extend the rules flexibly according to the noise characteristics of new domains.

## A.6 DETAILS OF REJECTING INVALID SAMPLES VIA LLMS

To eliminate invalid samples before functionality annotation, the AutoGUI pipeline prompts the annotating LLM to also determine the validity of samples by analyzing the predictability of the UI content changes. The LLM evaluates each sample against three criteria: 1) Explicitness of Changes: This measures how clearly the changes indicate the element's functionality. Changes that directly suggest functionality receive higher scores, while vague or irrelevant changes are not scored. 2) Relevance of Changes: This criterion assesses the significance of the modifications in relation to the element's intended function. Highly related modifications obtain a high score. No scores for irrelevant or unrelated content changes. 3) Predictability of Outcome: This involves determining how anticipated the interaction outcome is based on the changes, considering common web conventions and user experience principles. Highly predictable changes obtain a high score, whereas moderate, unexpected, or counter-intuitive outcomes receive no score.

Given the UI content changes as the input, the LLM first presents detailed reasoning processes about the three criteria and then outputs an overall score summing the individual scores for each criterion, with each contributing 0 to 3 points for a maximum of 9 points. The LLM presents three rejection results with temperature = 1.0 for each sample. Samples falling in the bottom 30% of average scores are considered invalid and discarded. This method ensures a balance between high recall of actual invalid samples and retention of valid samples. The prompt is shown in Tab. C, the rejection process is illustrated in Fig. G, and several representative rejection examples are shown in Fig. H. Note that UI content changes are represented as line-by-line differences in UI manipulation cases, and as descriptive changes in navigation scenarios.

To validate the effectiveness of the chosen score range 0-3, we test the ranges 0-2, 0-3, and 0-4 to select a range that helps to reduce false positives (valid but rejected) and increase true positives (invalid and rejected). We used 216 tasks, including 147 valid and 69 invalid samples as the test bed. We then drew a line chart illustrating the rejection ratios (Y-axis) for both valid and invalid samples against various threshold settings (X-axis) (Note that a sample whose score ranks below the threshold will be discarded). The selection criteria: the area under the curve (AUC) for the valid samples should be as small as possible, while the AUC for invalid samples should be large, ensuring valid samples rank higher. The results in Fig. F show that when using the score range 0-3, the AUC for invalid samples is the largest while the value for valid ones is small, which suggests that this range achieves a better tradeoff between retaining valid samples and rejecting as many invalid samples as possible.

## A.7 DETAILS OF LLM-BASED VERIFICATION

To improve the quality of functionality annotations, the AutoGUI pipeline prompts two LLMs (i.e.g, Llama-3-70B and Mistral-7B-Instruct-v0.2) as verifiers to assign scores to samples based on how well the target elements adhere to their functionality annotations. The LLMs receive as the input a) the target element along with its surrounding UI content (up to 20 lines), b) the functionality annotation of this element, and c) the outcome of interacting with the element, either being the UI line-by-line differences (at most 250 lines) in manipulation cases or the UI description after the interaction in navigation cases. Given these inputs, the two LLMs generate two responses containing a score. Samples that do not achieve two full scores are discarded for higher quality of the AutoGUI dataset. The used prompt is shown in Tab. D and an example is illustrated in Fig. I.

Table C: The rejection prompt used in the AutoGUI pipeline in UI manipulation cases. This example shows how the LLM assigns a low score to a sample that exhibits meaningless and unpredictable UI content changes.

---

(Requirements for rejection)

Your primary objective is to determine whether the changes in the webpage's content are sufficient for predicting the functionality of the webpage element causing these changes after being interacted with.

Instructions:

1. You will be shown the outcome (webpage changes) resulting from interacting with the element. The outcome can take one of two forms: changes to the webpage description, or line-by-line differences. For the latter form, here's what each prefix indicates:

Unchanged: Lines that are identical before and after the interaction.

Added: New lines that appear after the interaction.

Deleted: Lines that were present before the interaction but removed afterward.

Renaming: Lines indicating elements that were renamed due to the interaction.

Attribute Update: Lines showing elements whose attributes were updated during the interaction.

Repositioned: Elements that were moved to a different part of the webpage.

2. Analyze the provided outcome and provide detailed reasoning for whether this outcome helps to predict the element's functionality, considering the following stringent criteria:

1) Explicitness of Changes: Rate how directly the changes suggest the element's functionality. Score 1-3 for clear, unambiguous changes. Clearer changes obtain a higher score. No scores for vague, meaningless, or non-specific changes.

Positive Example: A button labeled "Show More" that, upon interaction, clearly adds new content below it. The direct addition of content clearly indicates a content expansion functionality. Score: 3

Negative Example: After clicking a "Details" button, the page layout changes subtly without adding relevant information or altering content in a meaningful way. The changes do not clearly relate to the button's presumed functionality. Score: 0

2) Relevance of Changes: Evaluate the significance of the modifications in relation to the element's intended function. Score 1-3 for changes that enhance understanding of the element's role. Highly related modifications obtain a high score. No scores for irrelevant or unrelated content changes.

Positive Example: Clicking on a "Contact Us" button opens a form to fill out, which is highly relevant to the button's intended functionality. Score: 3

Negative Example: Clicking on a "View Profile" link leads to a page refresh without displaying the profile or any related information, making the change irrelevant to the link's intended purpose. Score: 0

3) Predictability of Outcome: Assess how anticipated the interaction outcome is based on the changes, considering common web conventions and user experience principles. Score 1-3 for highly predictable outcomes. Highly predictable changes obtain a high score. No scores for outcomes that are moderate, unexpected, or counterintuitive.

Positive Example: Clicking or hovering over a "Help" icon reveals a tooltip with information, an outcome that users can easily predict given the icon's universal symbol for help. Score: 3

Negative Example: Hovering over a "Settings" icon changes its color but does not display any settings options, tooltips, or lead to a settings page, making the outcome unpredictable and the icon's purpose unclear. Score: 0

Provide a detailed analysis based on these criteria, ending with an overall score between 0-9.

4. Your output MUST follow this format:

Reasoning:

- Explicitness of Changes: [Detailed reasoning and score]

- Relevance of Changes: [Detailed reasoning and score]

- Predictability of Outcome: [Detailed reasoning and score]

Overall score: <score>[criterium 1 score] + [criterium 2 score] + [criterium 3 score] = [total score]</score> (output an additive equation enclosed with a pair of <score></score>)

Now it's your turn.

The interacted element is a <button> element named "Cancel" (interaction target)

Outcome:

(An example of interaction outcome)

After clicking a <button> element named "Cancel", the webpage exhibits the following variations:

Unchanged StaticText 'Email Address'

Unchanged textbox 'Email Address' required: True

Unchanged button 'Submit'

Unchanged StaticText 'Submit'

Before Attribute Update button 'Cancel'

After Attribute Update button 'Cancel' focused: True

Unchanged StaticText 'Cancel'

Unchanged contentinfo ''

Unchanged StaticText 'Your contact information is used to deliver requested updates or to access your subscriber preferences.'

Now please think step-by-step, examine the content variation in detail, and finally calculate scores according to the criteria. Do follow the output format "Reasoning: ... Overall score: <score>...</score>".

---

(LLM response)

Reasoning:

- Explicitness of Changes: The changes resulting from interacting with the "Cancel" button are mostly unchanged elements, except for the "Cancel" button itself, which now has a "focused: True" attribute. This change is not directly related to the button's functionality, as it does not provide clear insight into what the button does when clicked. Score: 1

- Relevance of Changes: The changes do not provide any significant information about the button's intended function. The button's name "Cancel" suggests that it might cancel some action or process, but the changes do not reveal what exactly is being canceled or how the button's functionality relates to the surrounding elements. Score: 1

- Predictability of Outcome: Given the name "Cancel" and the context of a form with an "Email Address" field and a "Submit" button, a user might expect the "Cancel" button to reset the form, clear the input field, or abort a submission process. However, the outcome does not provide any clear indication of what happens when the button is clicked. The changes are not predictable based on common web conventions and user experience principles. Score: 1

Overall score: <score>1 + 1 + 1 = 3</score>

Table D: The self-verification prompt used in the AutoGUI pipeline in UI manipulation cases. This example shows how the LLM assigns a low score to the incorrect functionality.

(Requirements for self-verification)
Given the following inputs:
1) Webpage content: This input represents the hierarchical structure of a webpage's elements, emphasizing semantic information and relationships. Each node in the tree includes details such as the element's role (e.g., button, link, heading), relevant attributes (e.g., expanded), and hierarchical relationships with other elements.
2) Task Description: This describes the action a user intends to perform (such as submitting a form, navigating to a particular section, or adjusting settings) or the information they seek (such as a specific content piece or form field). It also introduces a candidate element for evaluation and then presents the webpage changes caused by interacting with this element. Your task is to assess whether this element effectively facilitates the specified user action.
Your job is to:
1) Analyze the provided webpage content to understand the structure and semantics of the webpage's elements.
2) Evaluate the Candidate Element: Determine the suitability of the specified candidate element for the described action. Consider the element's role, attributes, and position within the hierarchy. Your evaluation should be grounded in how well these aspects align with the required functionality for the user's intended action.
3) Score the Element: Assign a score ranging from 0 to 3, enclosed within <score></score> tags. This score should reflect the degree to which the candidate element meets the action's requirements:
0: The element does not support the action in any capacity.
1: The element provides minimal support for the action.
2: The element supports the action but with limitations.
3: The element fully supports the action without significant limitations.
4) Provide Reasoning: Before presenting your score, offer a detailed explanation of your reasoning. This should cover your analysis of the webpage content, the relationship between the candidate element and the specified action, and how these factors informed your scoring decision.
5) Format for Your Answer:
Reasoning: (Provide a comprehensive analysis covering the webpage's insights, the relationship between the specified action and the candidate element, and the rationale behind your scoring decision.)
Score: <score>[$YourScoreHere$]</score>
(An in-context exemplar)
Example:
Webpage content:
[0] RootWebArea 'Rental Cars at Low, Affordable Rates'
[1] dialog 'Vehicle Class' modal: True
[2] radiogroup 'Vans'
[3] radio 'Minivans' checked: false
[4] radio 'Passenger Vans' checked: false
[5] radio 'Cargo Vans' checked: false
[6] button 'Cancel'
[7] button 'Apply Filter'
Task Description: Please identify the target element. The element helps users narrow down their vehicle choices to minivans specifically
Candidate element: [3] radio 'Minivans'
After interacting with the candidate element, the webpage exhibits these changes:
Upon clicking the "Minivans" <input> element, a new "Remove Filter" button is added to the dialog modal. The radio buttons for different vehicle classes remain unchanged, but the "Minivans" radio button is checked after the interaction.
Reference response:
Reasoning: The provided webpage content outlines a clear hierarchical structure for selecting vehicle types on a car rental webpage, categorized into sections like Cars, Vans, and Trucks, each with its own set of options represented as radio buttons.
The task involves narrowing vehicle choices to minivans. The candidate element is part of the 'Vans' radiogroup on a car rental webpage. This directly supports the user's action of narrowing choices to minivans. The element's interaction leads to its checking and introduces a "Remove Filter" button. Its role, position, and functionality support the user's intent.
Score: <score>3</score>
(The self-verification task)
Now it's your turn.
Current webpage content: (A UI content example)
[0] RootWebArea 'X for Business | X tips, tools, and best practices' focused: True
[2] link 'Get started in your ads account'
[4] link 'Campaign setup'
[6] link 'Editing and optimization'
[8] link 'Measurement'
[9] heading "Be what's happening"
[10] StaticText 'Spread your message, grow your audience, and get more traffic with Twitter Ads.'
=> [11] link 'Learn more' (The interaction target)
[13] heading 'Twitter is where people come to discover what's happening.'
[14] StaticText 'With over 500 million Tweets being sent each day, ...'
[15] StaticText 'Whether you're a CEO strengthening their personal brand...'
[16] link 'Get started with Twitter for Business'
[17] button 'Get started with Twitter for Business'
[18] heading "What's new on Twitter Business Outlook"
[20] link 'Google'
Task Description: Please identify the target element. This element redirects users to a non-existent page, providing a custom error message and alternative content instead of the expected information. (The functionality annotation of the element)
Candidate element: [11] link 'Learn more' (The interaction target)

(The interaction outcome)
After interacting with the candidate element, we navigate to a new webpage that contains these contents:
Region 1 (Header): Contains the website's logo and navigation links.
Region 2 (Error Message): Displays an error message indicating that the page does not exist, accompanied by a humorous image and a link to potentially related content.
Region 3 (Footer): Provides links to various policy and information pages, as well as copyright information.
Overall Functionality: This webpage serves as a custom 404 error page, providing a lighthearted and engaging way to notify users that the requested page is not available.

Now please do your job. Your answer:

(LLM response)
Reasoning: The candidate element, a link labeled 'Learn more', is part of the webpage's main content, surrounded by descriptions and headings related to Twitter for Business. However, upon interacting with this element, the webpage exhibits significant changes, redirecting users to a custom 404 error page with a humorous image and a link to potentially related content. In fact, the element's interaction leads to a completely different webpage with an error message and alternative content, which is unrelated to the initial webpage's purpose. The element's role, position, and functionality do not align with the user's intended action.
Score: <score>0</score>

Table E: Examples of the task templates used by the AutoGUI pipeline to convert element-functionality pairs to functionality grounding/referring tasks. {functionality} and {point} are place-holders to be replaced with the real functionalities and target points.

| Task Type | Examples |
|---|---|
| Grounding | For the element I describe in this page capture, can you predict their positions (with point)? {functionality} 
 In this web page image, please locate the element as I describe it (with point). {functionality} 
 Given a screenshot, I will describe a specific element; your task is to predict their locations (with point). {functionality} |
| Referring | Describe the function of the element at {point} on the screen. 
 Detail the functionality of the UI element positioned at {point}. 
 What kind of input or interaction is expected at the point marked {point}? |

## A.8 DETAILS OF GROUNDING/CAPTIONING TASK GENERATION

After collecting the element-functionality pairs, the AutoGUI pipeline converts these pairs into functionality grounding and captioning tasks by formatting a multitude of task templates (several examples are shown in Tab. E). A functionality grounding task requires a VLM to output point coordinates of the element fulfilling the given functionality, while a captioning task demands that the VLM articulate a functionality description for an element, given its coordinates. It is important to note that each element-functionality pair is utilized to generate both a grounding task and a captioning task.

To optimize training efficiency and minimize token expenditure, all point coordinates are normalized within the range $[0, 100]$. For tokenization, we employ the tokenizer from Qwen-VL-Chat without incorporating special tokens for the numerical range 0-99.

## B IMPLEMENTATION DETAILS

### B.1 HUMAN EVALUATION DETAILS

To justify the efficacy of the AutoGUI pipeline, we conducted a comparative evaluation of annotation correctness between a trained human annotator and the AutoGUI system. The human annotator was a student proficient in using digital devices, ensuring familiarity with diverse user interfaces.

We selected a set of 30 invalid samples, each showcasing a variety of element functionalities, to prepare the annotator for the annotation process. These functionalities included drop-down menu expansions, menu item selections, date-pickers, filtering options, pop-up modals, webpage navigation, and zooming in/out buttons. The purpose of this selection was to expose the annotator to a broad spectrum of potential UI interactions, enhancing their ability to accurately assess element functionality based on UI content changes.

During the training phase, we provided the annotator with detailed guidelines, including three specific criteria outlined in Fig 6, to ensure the clarity and correctness of their annotations. Additionally, we incorporated 15 invalid samples to instruct the human annotator on how to identify and exclude these cases during the evaluation process. These invalid samples encompassed scenarios such as incompletely loaded UIs, network failure incidents, login restrictions, and UIs displaying inappropriate content.

Following the training stage, the human annotator evaluated a total of 146 samples. Remarkably, the annotator successfully identified all invalid samples, achieving an overall annotation correctness rate of 95.5%. The few incorrect annotations were categorized as such due to vagueness or instances of hallucination, where the descriptions did not accurately reflect the UI elements.

### B.2 FINE-TUNING DETAILS

Qwen-VL-Chat Bai et al. (2023) and SliME Zhang et al. (2024b) are selected as the base models in the experiments. To investigate the scaling effects of our dataset, 25k, 125k, and the entirety of the 702k samples in the training split are used as training data in the three scaling experiments. For the first two smaller-scale experiments, a subset of the 702k data is randomly sampled.

Table F: The training hyper-parameters used for fine-tuning Qwen-VL in the experiments.

| Hyper-Parameter | Value |
| --- | --- |
| Epoch | 1 |
| Global batch size | 128 |
| #GPUs | 8 |
| Learning rate | 3e-5 |
| weight decay | 0.1 |
| ADAM Beta2 | 0.95 |
| Warm-up ratio | 0.01 |
| LR scheduler | Cosine |
| Model max length | 768 |
| LoRA | ViT + LLM |
| DeepSpeed | ZeRO-2 |
| #Parameters | Trainable params: 234,500,864
All params: 9,891,436,032
Trainable%: 2.3707 |
| Data type | BFloat16 |

Table G: The training hyper-parameters used for fine-tuning SliME in the experiments.

| Hyper-Parameter | Value |
| --- | --- |
| Epoch | 1 |
| Global batch size | 128 |
| #GPUs | 8 |
| Learning rate | 3e-5 |
| weight decay | 0.0 |
| ADAM Beta2 | 0.95 |
| Warm-up ratio | 0.03 |
| LR scheduler | Cosine |
| Model max length | 2048 |
| Frozen module | ViT |
| DeepSpeed | ZeRO-2 |
| #Parameters | Trainable params: 7535796224
All params: 8364644352
Trainable%: 90.09 |
| Data type | BFloat16 |

Pilot experiments find that the non-UI training data (i.e., LLaVA-instruct-150k and the Cauldron) significantly outnumber the 25k and 125k UI training data, resulting in data imbalance that biases the trained UI-VLM towards the general Q&A tasks in the non-UI data and leads to inferior UI grounding performance. To tackle this issue, the 25k/125k samples are resampled to the same number of the non-UI training data to enable the UI-VLM to acquire more supervising signals from the UI data. This resampling approach is not employed in the 702k experiment as this experiment does not encounter the imbalance issue.

We train our UI-VLM based on the HuggingFace Transformers[5] and the PEFT library[6]. The training configuration is shown in Tab. F and Tab. G.

## C  ADDITIONAL EXPERIMENTAL ANALYSIS

### C.1  GROWING GROUNDING PERFORMANCE BROUGHT BY SCALING DATA SIZE

To further investigate the benefit of scaling the AutoGUI functionality data, the histogram of distance from a predicted point to the ground truth box center is plotted for the 25k, 125k, and 702k experiments. The results in Fig. J demonstrate that the distance distributions become denser at

---

[5]https://huggingface.co/docs/transformers/index

[6]https://huggingface.co/docs/peft/index

lower ranges, suggesting that increasing the AutoGUI training data leads to consistently improved grounding performances.

## C.2 CASE ANALYSIS ON FUNCPRED TEST SPLIT

**Successful cases** Fig. K demonstrates several examples of the grounding results from Qwen-VL trained with the 25k, 125k, and 702k AutoGUI data. The model trained with the 702k data (ours-702k) exhibits more accurate functionality grounding performance. For instance, Fig. K (a) shows that ours-702k predicts the point right on the target (The 'Get an account' button) while the other two models slightly miss the target. Case (c) shows that ours-702k correctly understands the functional intent to locate the WordPress logo, in contrast to the other models, which incorrectly focus on the text 'Get WordPress'. Additionally, case (f) illustrates that ours-702k successfully locates the three-dot menu icon, aligning with the intent to expand a dropdown menu. These results suggest that increasing the AutoGUI training data enhances the model's ability to understand complex functional intents and to recognize diverse iconic elements accurately.

**Failure cases** To explore the limitations of our model, we analyze several failure cases across the scaling experiments, as shown in Fig. L. The primary failure cases comprise (1) Difficulty in accurately locating very small target elements, as illustrated by the tiny 'Policy' button in case (a); (2) Misunderstanding functional intents, as shown in case (b) where the three models fail to locate the element for account creation and case (g) where ours-702k mistakenly focuses on navigating to previous content instead of subsequent content; (3) Challenges in recognizing abstract iconic elements, as seen with the map style icon in case (d) and the compass icon in case (f).

Despite these challenges, the enhanced performance observed with ours-702k supports the potential of the AutoGUI pipeline to further improve functionality grounding. The successful cases underscore that increasing the size of the training dataset not only boosts the model's ability to interpret functional intents but also its capability to process a variety of textual and iconic elements effectively.

## C.3 CASE ANALYSIS ON MOTIF TEST SPLIT

We evaluate the instruction following ability on MoTIF dataset. Our analysis focuses on two aspects: (1) what improvements our model can achieve with the scaling of our functionality dataset (Fig. M); and (2) in which scenarios our model still fails to achieve correct grounding (Fig. N).

Fig. M shows that the model can more accurately understand the action instruction and make meaningful localization as scaling improves from 125k to 702k. For instance, when the objective is to *click sleep noise recording and click enable*, the model can comprehend the semantics of this global objective and identify *turn on*. Additionally, the model can mitigate localization errors, such as the 702k being more accurately positioned on the target element (e.g., the icon of *reservation*) than the 125k. However, MoTIF still struggles with certain tasks. For example, as shown Fig. N, it has difficulty with localization in fine-grained steps for the instruction *search for Kingston Drive and show me the route to it*. It can be seen that the model does not effectively understand situations involving widget pop-ups (e.g., protocol and advertisement). This may be attributed to the weak semantic connection between pop-ups and the instruction. Furthermore, the model still falls short in precise localization. Enriching the dataset further could alleviate this issue.

## D LIMITATIONS

AutoGUI is dedicated to providing an autonomous way to collect scalable UI grounding/captioning data for training capable UI-VLMs. However, AutoGUI still encounters several limitations:

**Lack of Diverse Mobile App Data.** As many Apps implement anti-emulator code, it is extremely difficult to navigate through popular Apps, such as TikTok and WeChat, on Android emulators. To circumvent this issue, AutoGUI renders webpages at various resolutions, including smartphone resolution, to mimic diverse device types. Although mainstream websites, such as YouTube and Reddit, provide delicately designed webpage responsiveness for various resolutions, a number of less common websites do not possess such flexible responsiveness and distort severely when rendered at smartphone resolutions. Therefore, collecting UI data at a smartphone resolution probably leads to domain gaps between the collected data and real smartphone Apps that are not rendered with HTML.

**AutoGUI is Not Indented to Record Task-Oriented Interaction Trajectories.** AutoGUI randomly interacts with UIs to record transition trajectories and utilize the UI content changes to predict the functionalities of the interacted elements. Hence, the collected trajectories do not provide high-level task semantics. In other words, the AutoGUI dataset does not contain tasks that combine multiple low-level steps, such as selecting a check-in date and then a check-out date. These long-horizon tasks are usually generated by human annotators in the existing works Deng et al. (2024); Rawles et al. (2023). In future work, we can also utilize capable LLMs to generate high-level tasks and then prompt the LLMs to interact with UIs according to the tasks.

**AutoGUI Cannot Annotate UI Elements That Modify Content on the Internet** To avoid causing potential contamination on the Internet and bearing unexpected responsibilities, we try our best to eliminate interaction samples that manipulate sensitive elements that probably modify contents on the Internet. For example, elements used to post comments, make purchases, and enter account information are discarded. Consequently, the AutoGUI pipeline mainly annotates elements that only support read-only functionalities.

## E  POTENTIAL SOCIETAL IMPACT

The potential societal impacts of the proposed AutoGUI can be considered across various dimensions:

**Accessibility Enhancements** VLMs trained with the AutoGUI data obtain stronger UI grounding capabilities, thereby possessing the potential to act as UI agents. By enabling context-aware understanding of UI functionalities, the VLMs can help users locate elements on complex UIs, significantly improving accessibility features in software. This could lead to the development of applications that are more intuitive for users with disabilities, such as those requiring screen readers or other assistive technologies.

**Research Impact**: By reducing the labor and time required for annotating UI data via the AutoGUI, the industry and academia could lower costs to easily build UI agents. This could also shift labor demands towards more creative and strategic roles rather than repetitive annotation tasks.

**Privacy and Security Concerns**: Although we employ precautions of eliminating samples related to sensitive UI elements (e.g., avoid interacting with elements modifying the Internet and use only popular public websites without exposing privacy), corner cases still exist on the vast Internet. UI data involving either content modification or personal information are hard to discern as UI designs are distinct and no universal detection rules exist. Therefore, it is essential for cyber-security research to consider the potential leakage of personal information in the collected data and devise preemptive protective approaches.

**Potential for Bias and Fairness**: The bias of the LLMs used in the AutoGUI annotation pipeline is probably reflected in the collected data, leading to a trained UI-VLM that inherits the bias. Therefore, mitigating bias in the LLM's annotations will be important for developing fair VLM agents that align with the values of users from diverse cultures.

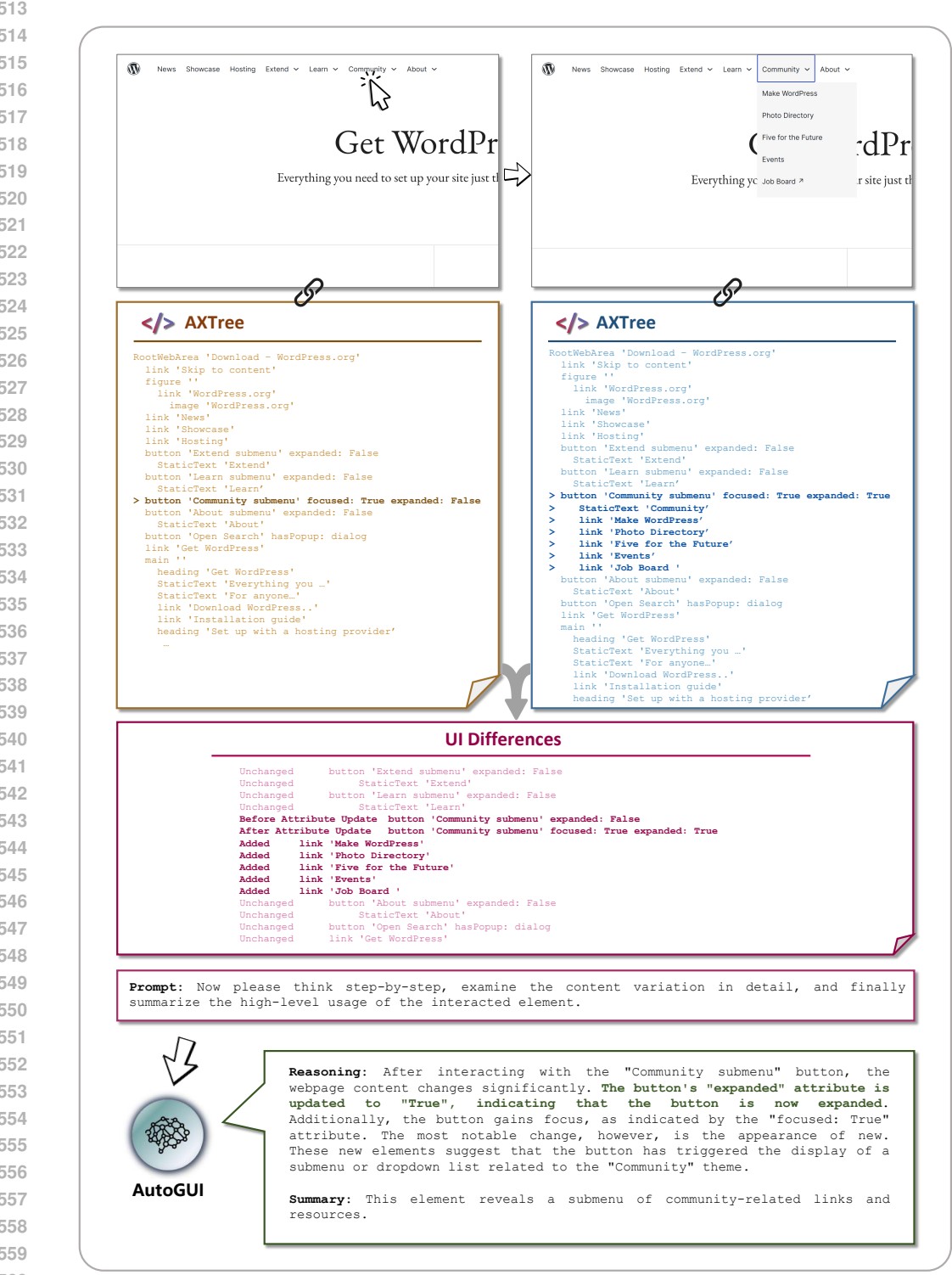

Figure D: An example of the AutoGUI functionality annotation using UI AXTree differences. AutoGUI records the AXTrees before and after interaction and then generates line-by-line differences with our custom change markers. Subsequently, the LLM takes the differences as input to predict the element functionality.

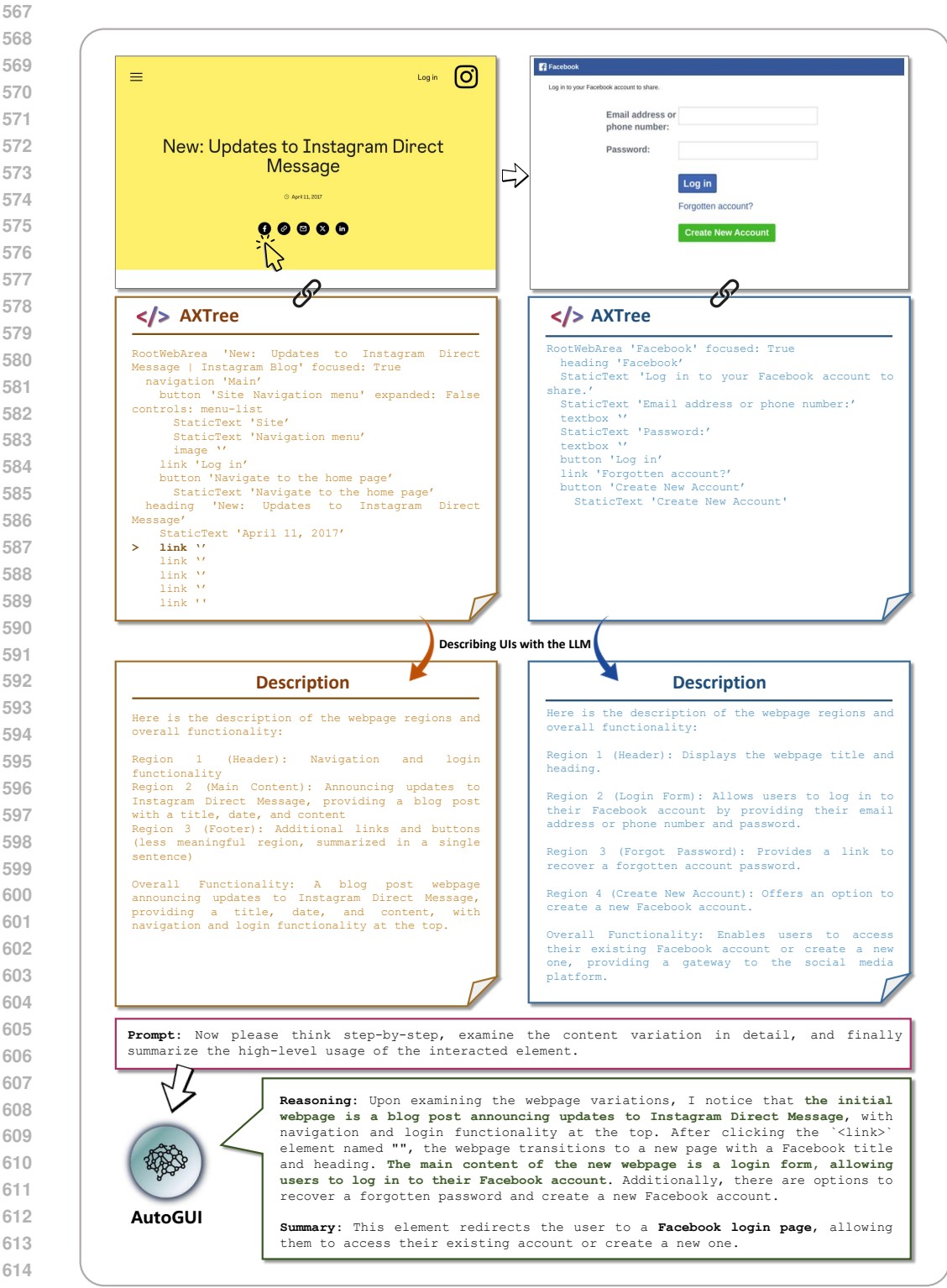

Figure E: An example of the AutoGUI functionality annotation using UI descriptions. AutoGUI records the AXTrees before and after interaction and then prompts the LLM to describe the AXTrees in detail. Subsequently, the LLM takes the two descriptions as input to predict the element functionality.

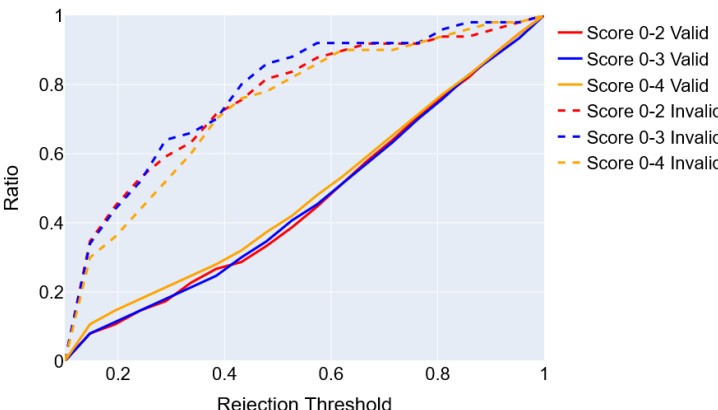

Figure F: **Rejection outcomes of the three score ranges used by the LLM-based rejector**. We plot the ratios of rejected valid and invalid samples when setting the rejection threshold to different values from 0.0 to 1.0. As the rejection threshold increases, more invalid and valid samples will be rejected. We expect the area under the curve (AUC) for rejecting invalid samples to be as large as possible while the AUC for rejecting valid ones to be as small as possible. The figure shows that using a score range of 0-3 leads to the largest AUC for invalid samples and a small AUC for valid ones.

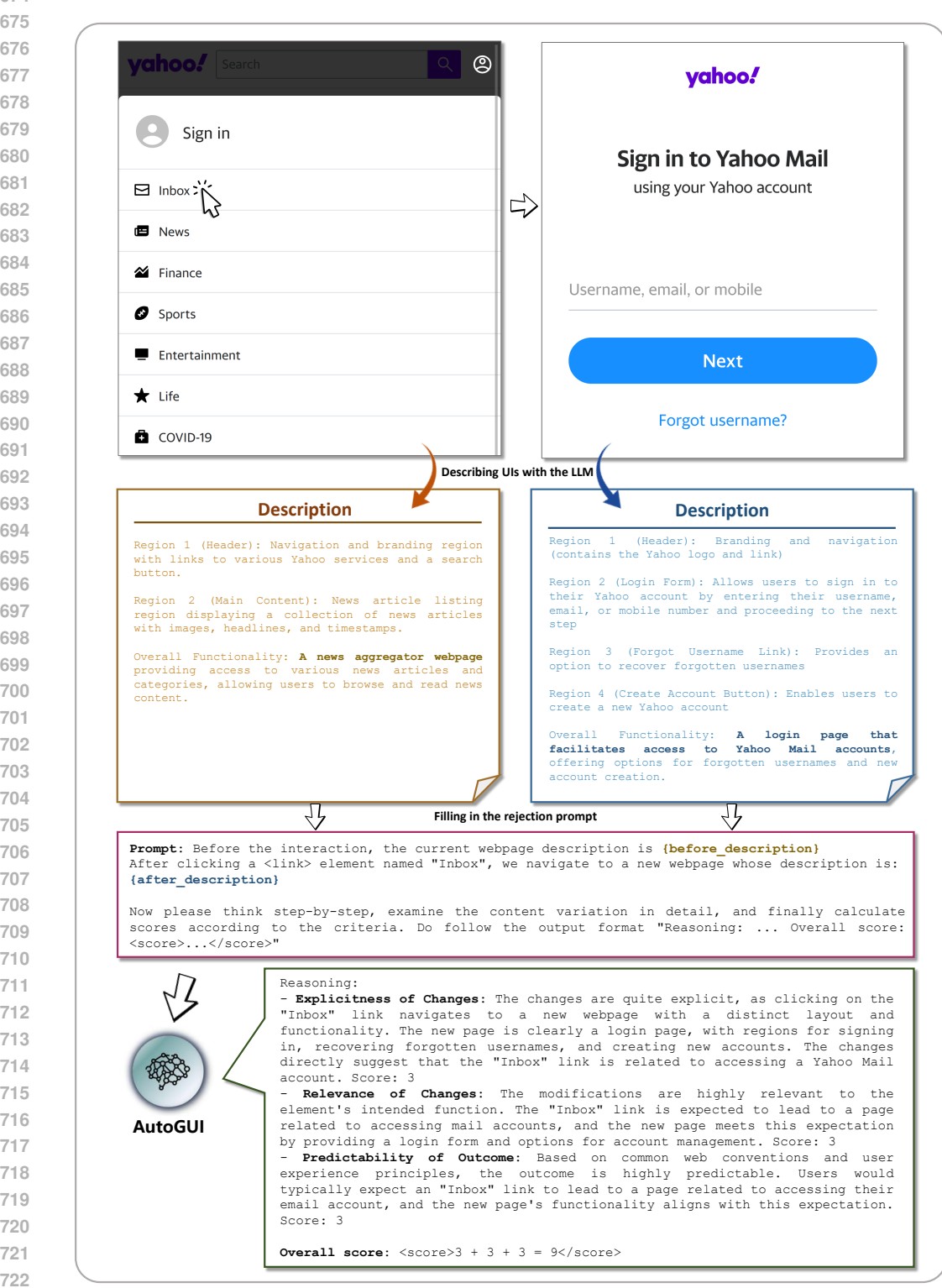

Figure G: An example of AutoGUI prompting the LLM as a rejector to determine whether a sample shows meaningful UI content changes sufficient for predicting the functionality of the interacted element. The sample shown is a navigation case in which AutoGUI uses UI descriptions, instead of line-by-line differences, to make decisions.

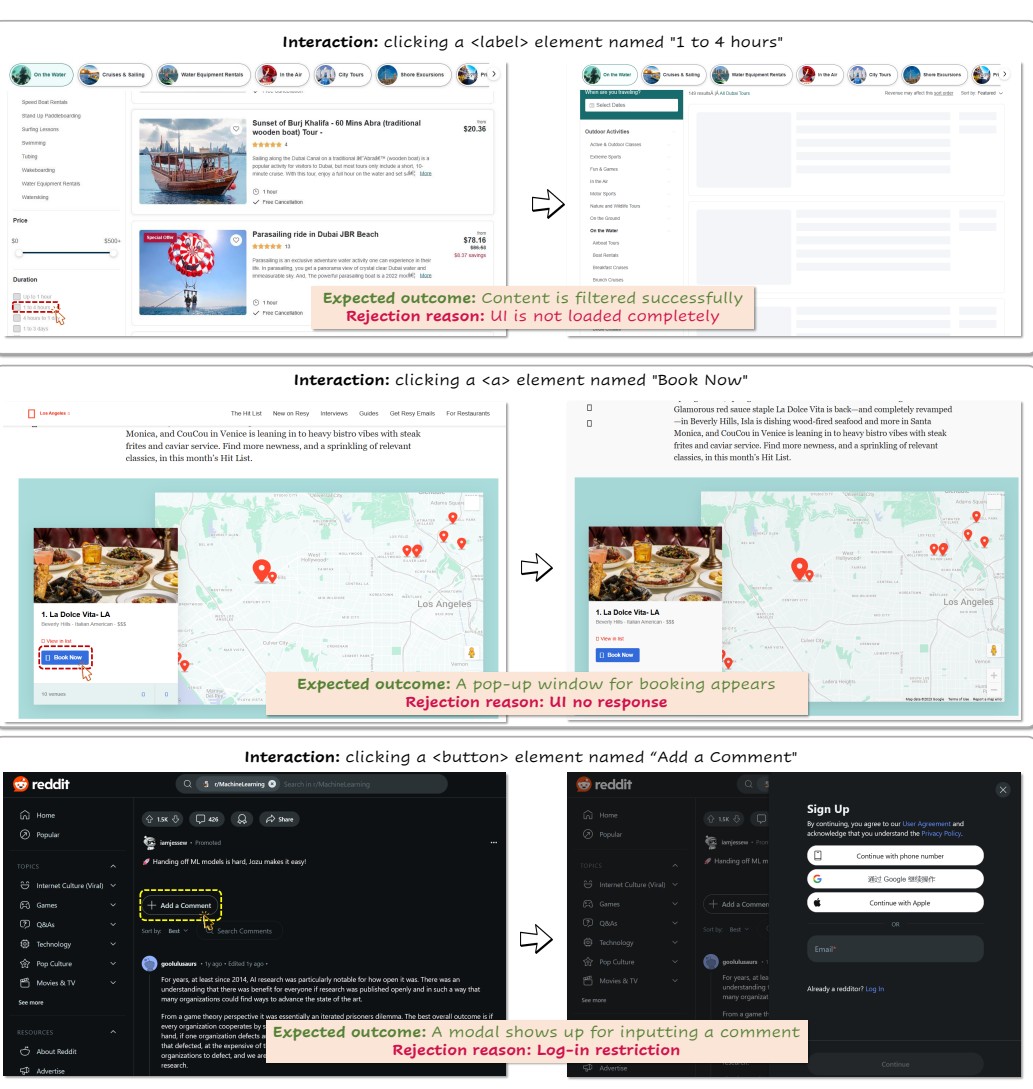

Figure H: **Examples of samples rejected by the AutoGUI pipeline.** The first sample encounters incompletely loaded content that interferes LLM annotation. The second encounters a no-response issue where the pop-up window fails to appear. The third shows a case where an unexpected log-in page pops up to interrupt the functionality of the "Add a Comment" element.

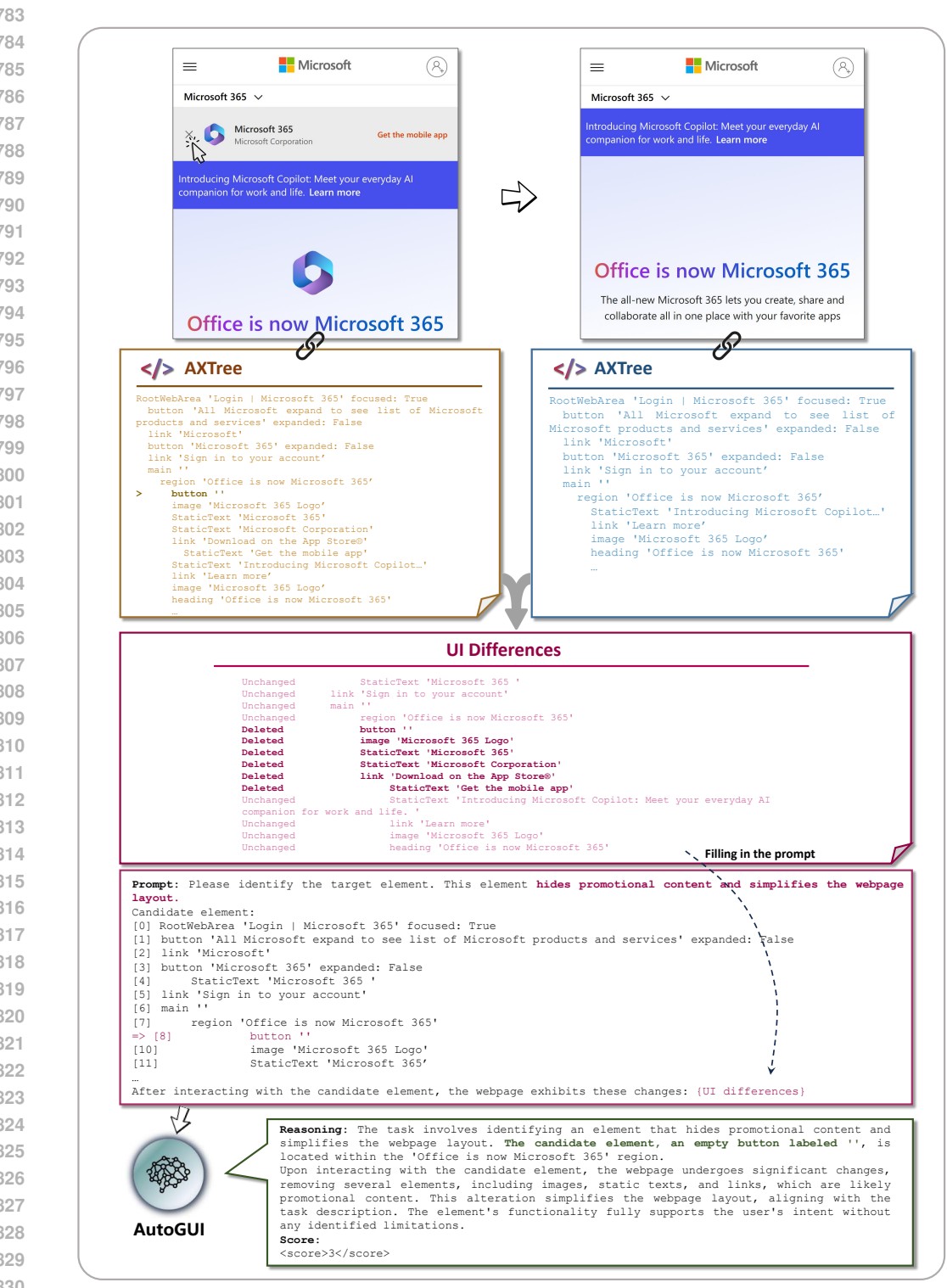

Figure I: An example of AutoGUI prompting the LLM as a self-verifier to determine whether an element supports its functionality annotation. The sample shown is a manipulation case in which AutoGUI uses UI line-by-line differences to make decisions about whether a button fulfills the intent of hiding promotional content.

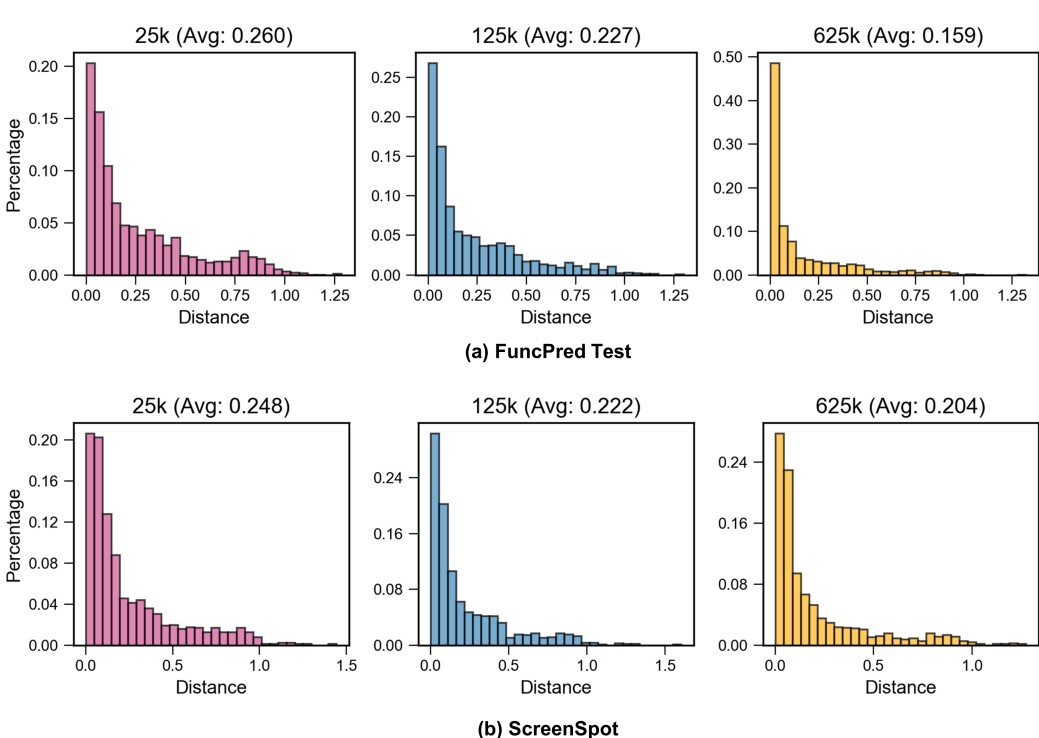

Figure J: **Histograms of distances from predicted points to ground truth box centers.** The distance from the normalized coordinate of a predicted point to its corresponding GT box center is calculated for all samples. Then, the histograms of these distances are illustrated to demonstrate the growing grounding performances brought by scaling the AutoGUI data size. The averaged distance for each experiment is displayed on the subplot title.

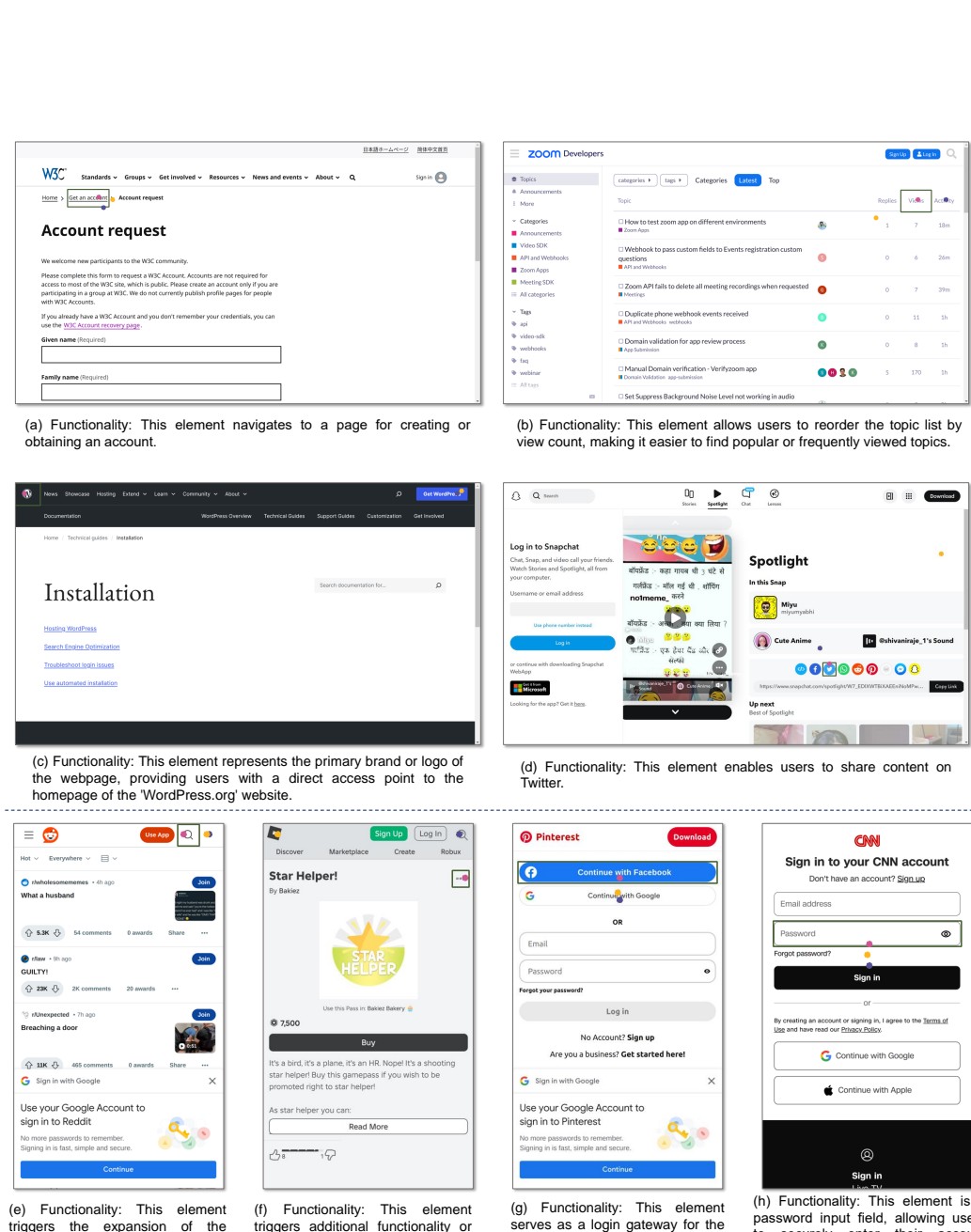

(a) Functionality: This element navigates to a page for creating or obtaining an account.

(b) Functionality: This element allows users to reorder the topic list by view count, making it easier to find popular or frequently viewed topics.

(c) Functionality: This element represents the primary brand or logo of the webpage, providing users with a direct access point to the homepage of the 'WordPress.org' website.

(d) Functionality: This element enables users to share content on Twitter.

(e) Functionality: This element triggers the expansion of the search functionality on the webpage, allowing users to access more extensive search options.

(f) Functionality: This element triggers additional functionality or navigation within the webpage, such as revealing a dropdown menu.

(g) Functionality: This element serves as a login gateway for the Pinterest app, allowing users to authenticate their accounts using Facebook.

(h) Functionality: This element is a password input field, allowing users to securely enter their account password for authentication during the login process on the CNN website.

Figure K: **Visualization of the successful functionality grounding examples for ours-625k.** The ground truth bounding boxes, ours-625k predictions, ours-125k predictions, and ours-25k predictions are drawn in green, pink, blue, and orange, respectively.

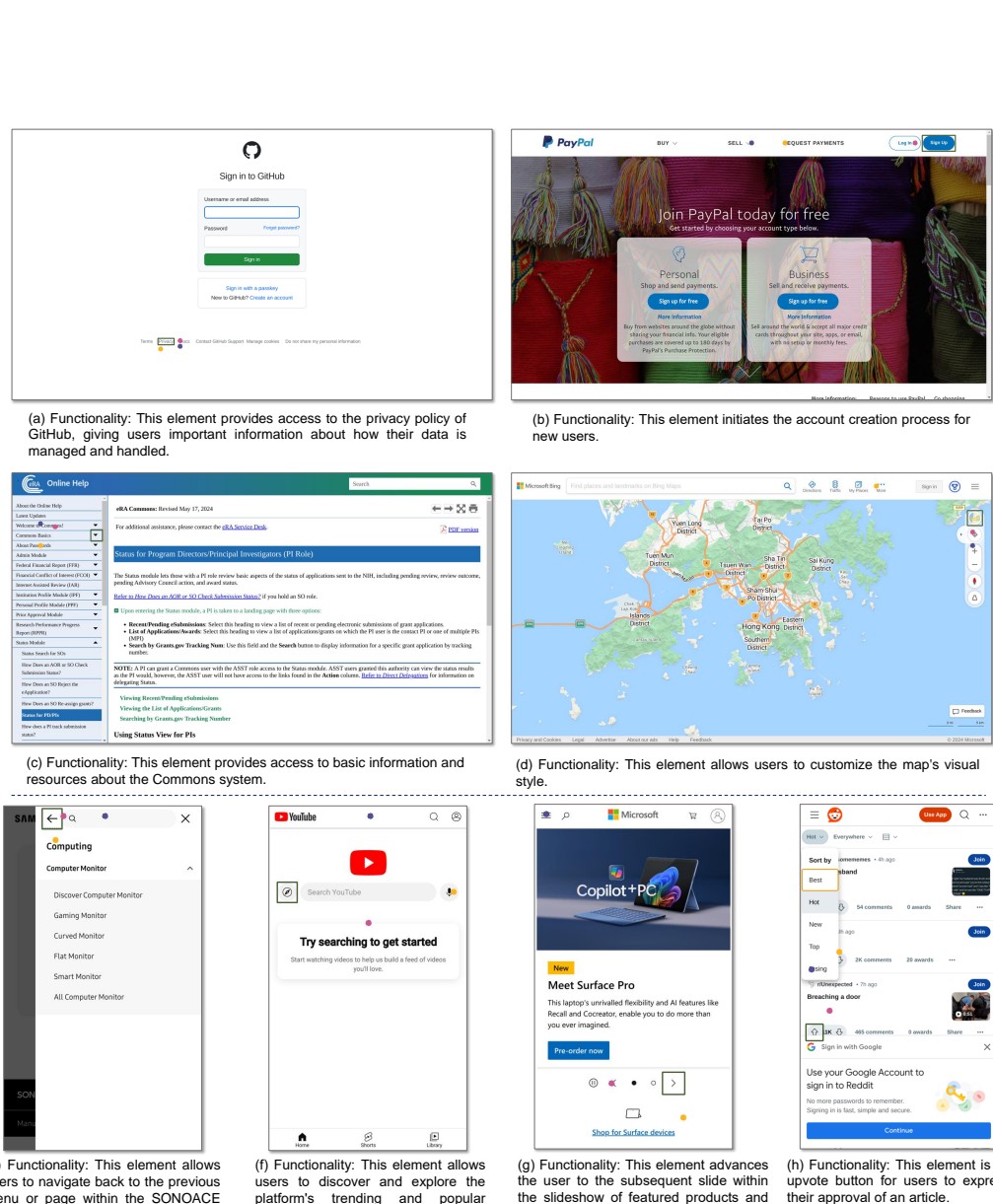

(a) Functionality: This element provides access to the privacy policy of GitHub, giving users important information about how their data is managed and handled.

(b) Functionality: This element initiates the account creation process for new users.

(c) Functionality: This element provides access to basic information and resources about the Commons system.

(d) Functionality: This element allows users to customize the map's visual style.

(e) Functionality: This element allows users to navigate back to the previous menu or page within the SONOACE R3 | Samsung Support Bangladesh webpage.

(f) Functionality: This element allows users to discover and explore the platform's trending and popular content, providing a gateway to various sections and categories of the video-sharing platform.

(g) Functionality: This element advances the user to the subsequent slide within the slideshow of featured products and announcements, providing a means for users to browse through the displayed content.

(h) Functionality: This element is an upvote button for users to express their approval of an article.

Figure L: **Visualization of failure examples in the scaling experiments.** The ground truth bounding boxes, ours-625k predictions, ours-125k predictions, and ours-25k predictions are drawn in green, pink, blue, and orange, respectively.

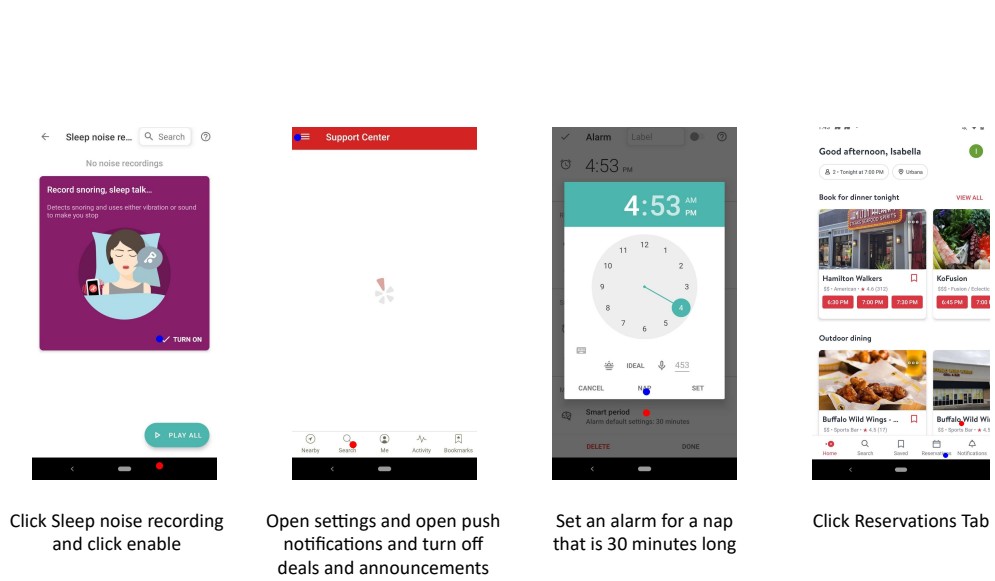

Click Sleep noise recording
and click enable

Open settings and open push
notifications and turn off
deals and announcements

Set an alarm for a nap
that is 30 minutes long

Click Reservations Tab

Figure M: Evaluation results of the model trained on 625k (blue dot) and 125k (red dot).

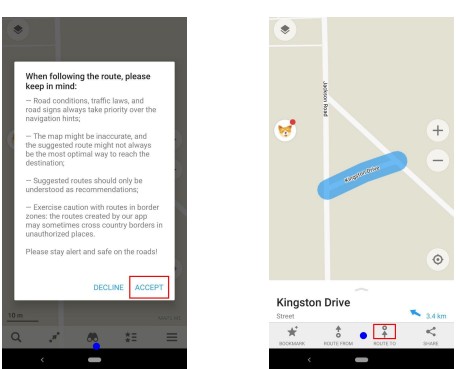

Search for Kingston Drive and show me the route to it

Figure N: Bad cases on MoTIF.

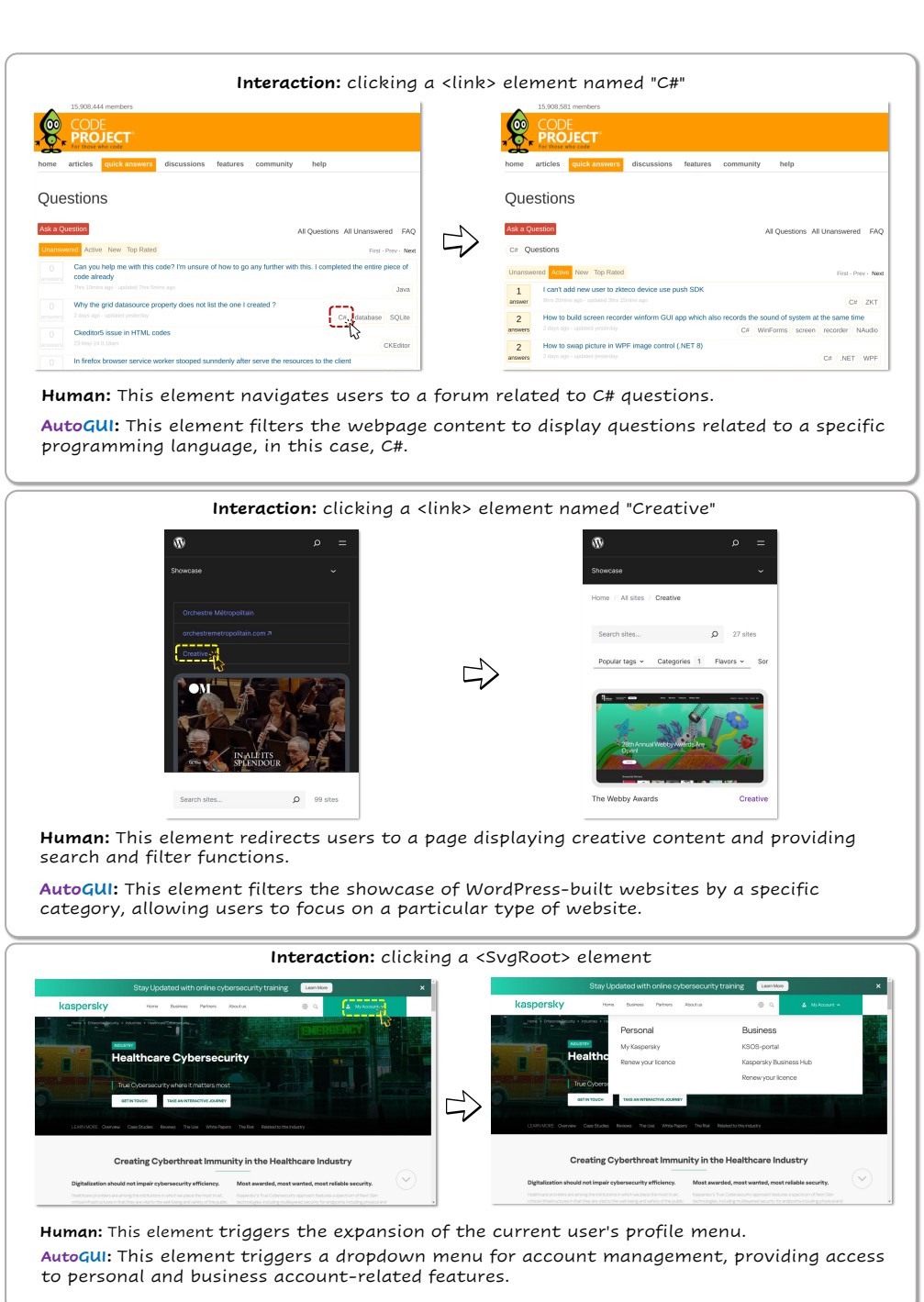

Figure O: Comparing the annotations generated by a trained human annotator and the proposed AutoGUI pipeline. We can see that AutoGUI annotations are more detailed and clear than those by the human annotator.

