# OpenReview forum: "AutoGUI: Scaling GUI Grounding with Automatic Functionality Annotations from LLMs"
_ICLR.cc/2025/Conference — Submitted to ICLR 2025_

### Official Review · Reviewer_vvNy · 2024-10-26

**Soundness:** 2
**Presentation:** 3
**Contribution:** 2
**Rating:** 3
**Confidence:** 4

**Summary:**

The paper introduces AutoGUI, an automatic annotation pipeline designed to scale the grounding of GUI elements by leveraging LLMs' annotation. The authors address the limitations of existing UI datasets, which either lack contextual functional descriptions or are limited in scale. The proposed AutoGUI pipeline automatically collects UI interaction trajectories and uses LLMs to annotate UI elements based on the observed changes in UI content before and after an interaction. For annotation quality, the authors implement LLM-aided rejection and verification mechanisms to filter out invalid or incorrect annotations w.o. human intervention.

**Strengths:**

1. Addressing the data scarcity in UI understanding is highly significant for advancing VLMs in GUI-oriented applications. The ability to automatically generate large-scale, high-quality GUI datasets can accelerate research and development.
2. The AutoGUI-704k provides 704,000 high-quality samples featuring multi-resolution, multi-device screenshots, diverse domains, and detailed functionality annotations. It demonstrate practical applications.
3. Experiments show that models trained on the AutoGUI-704k dataset significantly enhance UI grounding capabilities of VLMs, exhibit scaling effects with increased data size, and outperform models trained on existing (manual) GUI datasets.

**Weaknesses:**

My concerns about this paper mainly revolve around the following three points:

1. **Evaluation on the FuncPred Test Set:** The authors state that the FD metric ("FuncPred is the test split from our collected functionality dataset. This benchmark requires a model to locate the element specified by its functionality description") is essentially a test set drawn from the same format as their training data. It is expected that scaling up their dataset shows effectiveness on this test set. Additionally, further improvement in performance after continued training with the SeeClick data is also reasonable. However, on the **VisualWebBench**, adding the SeeClick data for continued training results in a performance gain that surpasses that achieved with the authors' full dataset:
    - **Data Quality Concerns:** Since the SeeClick data is of the type focusing purely on element grounding (e.g., ScreenSpot), does this partially reflect that the data generated by the proposed method may be of lower quality?
    - **Performance Drop on RefExp:** Moreover, after training, the performance on the RefExp benchmark shows inferior performance for unknown reasons, yielding lower results than training with only SeeClick data. Can the authors provide explanations for this unexpected drop?
2. **Performance Fluctuations with Finetuned Qwen-VL (Tab. 4):** In the main exp, regarding the experiments with finetuned Qwen-VL, it can be observed from Qwen-VL-AutoGUI702k that when finetuned using AutoGUI702k, there is significant performance fluctuation compared to the SeeClick baseline, with a range reaching up to ±20%. Do the authors have further explanations into these substantial performance variations?
3. **Unclear Process in Removing Invalid Samples:**
    - **Insufficient Explanation of Hand-Written Rules:** Authors mentions "hand-written rules" used in the process of removing invalid samples, but these rules are not well explained or detailed.
    - **Lack of Justification for LLM Predictability Scores:** The process of obtaining predictability scores from the LLM outputs lacks sufficient rationale. For instance, why does the scoring range from 0 to 3? More explanation is needed on how this scoring system handles errors, biases, and ambiguous cases.

**Questions:**

My assessment is primarily based on the solidity of the experiments in this paper. I'll try to adjust my assessment after reading the rebuttal.

---

> ### Author Response · Authors · 2024-11-22
> **Response to Reviewer vvNy (Part 2)**
>
> > Q3: When fine-tuned using AutoGUI702k, there is significant performance fluctuation compared to SeeClick. Do the authors have further explanations for these variations?
>
> We understand your concern and would like to further clarify it: the variations possibly originate from the discrepancy between the grounding tasks of the training data and benchmarks.
>
> Qwen-VL-AutoGUI702k is fine-tuned with a focus on functionality semantics, which improves its performance on benchmarks that require locating elements based on functional referring expressions (e.g., FuncPred, VWB AG, and MOTIF). However, it may achieve marginal improvements on benchmarks that rely more on element properties such as relative positions or text (e.g., RefExp, ScreenSpot).
>
> To make this explanation clearer, we inspect samples of each benchmark on which the performance gap between Qwen-VL-AutoGUI702k and SeeClick is noticeable:
>
> (1) Inspecting 50 samples on **FuncPred**, SeeClick underperforms due to:
>
> - (a) **Heavily rely on element texts (64%)**: SeeClick can locate an element correctly when the referring expression (RE) includes the displayed text of the target element but struggles when the expression does not. For example, when the RE explicitly mentions “log-in” for a “Log-in” button, SeeClick can correctly locate the element; interestingly, when the referring expression says “This element triggers the display of a password recovery page”, SeeClick mistakenly locates a “Password” text field instead of the “Forget password” button below the text field.
>
> - (b) **Almost hit the target (16%)**: The predicted point of SeeClick is very close to the ground truth bounding box.
>
> - (c) **Others (20%)**: locate either wrong elements or non-interactable areas without any explicit reasons.
>
> (2) Inspecting 20 samples on **VWB AG**, SeeClick is slightly worse due to:
>
> - (a) **Almost hitting the target (30%)**.
>
> - (b) **Heavily rely on element texts (30%)**. For example, given a referring expression “change the country/region of Calendar”, SeeClick mistakenly points to a calendar element instead of the “Country selection” menu.
>
> - (c) **Others (40%)**.
>
> (3) Inspecting 30 samples on **ScreenSpot**, Qwen-VL-AutoGUI702k underperforms due to:
>
> - (a) **Incorrect text localization (46.7%)**. ScreenSpot contains many text localization tasks. For example, it misidentifies "number 5" as "number 7" in a calculator UI.
>
> - (b) **Almost hit the target (40.0%).**
>
> - (c) **Others (13.3%).**
>
> (4) Inspecting 30 samples on **RefExp**, Qwen-VL-AutoGUI702k underperforms due to:
>
> - (a) **Difficulty with position-based REs (43.3%)**. The RefExp REs mostly are short phrases with position descriptions, such as “click on the third image on the right from the top”. As our AutoGUI dataset mainly focuses on element functionality, Qwen-VL-AutoGUI702k is not quite capable of understanding positional relationships. For example, a task says “select the symbol which is immediate to the left of keep me signed in”, but Qwen-VL-AutoGUI702k taps on the text “keep me signed in” instead of the checkbox left to this text.
>
> - (b) **Almost hit the target (36.7%).**
>
> - (c) **Others (20%).**
>
> (5) Inspecting 30 samples on **MOTIF**, SeeClick underperforms due to:
>
> - (a) **Failed to understand functionality (40%)**, such as "set a daily reminder to write my diary," due to a lack of functional data in its training set.
> - (b) **Almost hit the target (16.7%).**
>
> - (c) **Others (43.3%).**
>
> _Hope this detailed analysis can address your concern._

---

> ### Author Response · Authors · 2024-11-22
> **Response to Reviewer vvNy (Part 3)**
>
> > Q4: Why does the scoring range from 0 to 3? More explanation is needed on how this scoring system handles errors, biases, and ambiguous cases
>
> A: Thanks for raising this point. As explained in Section A.5, the 0-3 scoring range was chosen based on empirical findings that it strikes a **balance between annotation efficiency and data quality**. This range effectively filters out invalid samples while retaining most valid ones.
>
> In our preliminary experiments, we observed that a binary score (0/1) for the LLM rejector was **biased**, often undesirably assigning a score of 1 to ambiguous samples. This was because the LLM tended to treat minor annotation errors as having negligible impact on overall quality. Similarly, the LLM-based verifier exhibited this behavior, resulting in a higher proportion of incorrect and ambiguous annotations.
>
> (1) For the LLM rejector, to identify the optimal scoring range, we tested several thresholds: 0-2, 0-3, and 0-4. We conducted this evaluation on a set of 216 tasks, consisting of 147 valid and 69 invalid samples. We then plotted a rejection ratio curve for both valid and invalid samples at various thresholds (i.e., samples with scores below the threshold are rejected). The goal was to **minimize the area under the curve (AUC)** for valid samples, while maximizing it for invalid ones, ensuring that valid samples are ranked higher than invalid ones. Please refer to **Figure F** in the revised paper for a visual comparison.
>
> The results showed that the 0-3 range provided the best separation: it had the largest AUC for invalid samples and a small AUC for valid ones, indicating a better balance. This range was therefore selected for the LLM rejector.
>
> (2) For the LLM-based verifier, we tested the **verification accuracy** using 100 samples across the three scoring ranges: 0-2, 0-3, and 0-4. The verification accuracy is defined as the number of correctly classified annotations divided by the total (note that an annotation whose verification score is not full will be classified as incorrect). The 0-3 range achieved the highest accuracy at 96%, compared to 92% for 0-2 and 94% for 0-4. Further analysis revealed that the 0-2 range was prone to assigning high scores to ambiguous samples, reducing accuracy, while the 0-4 range was too stringent, misclassifying more valid annotations as invalid. Thus, the 0-3 range with higher accuracy was selected.
>
> In conclusion, the 0-3 scoring range was chosen to mitigate LLM bias in binary scoring and to effectively balance the retention of valid samples with the rejection of invalid ones.
> ***
> _We sincerely appreciate your detailed feedback. We hope the above response can address all your concerns. If you have any questions, we are pleased to provide further clarification!_

---

> ### Author Response · Authors · 2024-11-30
> **Official Comment by Authors**
>
> Dear Reviewer vvNy:
>
> We sincerely thank you for these constructive comments and evaluation of our paper. With the ICLR public discussion phase ending in two days, we kindly ask you to take a look at our responses. Our rebuttal provided more clarification of our framework and additional experiments in response to your concerns. Please let us know whether our response addresses your concerns or whether there is any further detail we can provide to help address these concerns.
>
> Thank you again for dedicating your time to reviewing our paper.

---

> > ### Author Response · Authors · 2024-12-04
> >
> > Dear Reviewer vvNy,
> >
> > As the discussion phase nears its conclusion, we would like to take this opportunity to express our sincere gratitude for your valuable feedback and constructive engagement throughout this process.
> >
> > With only a few hours remaining, we would like to ensure that all your concerns have been thoroughly addressed. In the latest response, we made a dedicated effort to comprehensively address your concerns, including those regarding **Dataset Effectiveness**, **Performance Gap Analysis**, and **Scoring Approach Rationale**. If you have the chance, we would greatly appreciate it if you could review our latest response to confirm whether it fully addresses your concerns.
> >
> > We deeply appreciate the time and thoughtfulness you have invested in this discussion and are eager to hear any further thoughts or suggestions you may have. Thank you once again for your support and attention to our work!
> >
> > Best regards,
> >
> > The Authors

---

### Official Review · Reviewer_ZWrk · 2024-10-29

**Soundness:** 1
**Presentation:** 3
**Contribution:** 2
**Rating:** 3
**Confidence:** 5

**Summary:**

This paper proposes a scalable and automatic UI data annotation pipeline that annotates GUI elements with detailed functionality descriptions. It also proposes a series of filters to improve the annotation quality including hand-written rules and LLM-based rejectors and verifiers. The paper shows that the collected annotations achieves high correctness comparable to the trained human annotator, thus reducing the burden of collecting GUI data. The experiments on the collected AutoGUI dataset show its effectiveness in GUI grounding task.

**Strengths:**

Quality & Clarity

The paper points out the shortcomings of current GUI datasets and proposes a data collection pipeline to address them. The collected data is first analyzed for correctness to ensure its quality, and its effectiveness is subsequently demonstrated through experiments. The writing is logically structured and clearly expressed.

**Weaknesses:**

Limited Evidence: The experiments cannot fully demonstrate the effectiveness of AutoGUI data.

1. This paper evaluates AutoGUI data on 6 benchmarks as shown in Table 4. The effectiveness of AutoGUI data can be assessed by comparing the results of *Qwen-VL-AutoGUI702k* and *SeeClick* as they use the same base model. The results on *FuncPred* benchmark are excluded from consideration as *FuncPred* is derived from AutoGUI dataset and shares the same data distribution with it. In the remaining 5 benchmarks, *Qwen-VL-AutoGUI702k* performed worse than *SeeClick* in 3 of them (VWB EG, RefExp, and ScreenSpot). The paper attributes this performance gap to the absence of data from Apple devices and desktop software in the AutoGUI dataset. However, the *ScreenSpot* benchmark has 3 subsets including Web, Desktop and Mobile, and there is a lack of experiments on the Web subset in *ScreenSpot* to support this argument.
In summary, the existing experiments cannot prove the effectiveness of AutoGUI training data.

2. Also the Table 4:
  a) By comparing the results of *Qwen-VL-AutoGUI702k* and *Qwen-VL-AutoGUI702k\**, it is observed that the introduction of SeeClick training data improves the performance of Qwen-VL on all benchmarks;
  b) By comparing the results of *SeeClick* and *Qwen-VL-AutoGUI702k*, it is observed that the introduction of AutoGUI data reduces the performance of Qwen-VL on *RefExp*, and did not significantly improve the performance on *ScreenSpot*.
These 2 results indicate that the role of AutoGUI data is not as significant as SeeClick training data.

3. The paper identifies 3 annotation types in existing UI grounding training data (see Fig.2), but only two of them (Brief Function & HTML Code) are chosen in the experiments (see Table 5). An additional experiment on the Visual appearance and category annotation type would provide a more complete demonstration.

Limited Significance
1. This paper mentions that the ultimate goal of the research is to enable next-generation software automation. However, the work presented here focuses solely on the GUI grounding task, lacking exploration of practical application scenarios.
2. The experimental results indicate that the proposed AutoGUI dataset has not led to substantial advancements in GUI grounding task compared to previous work. (As shown in Table 4, apart from the self-built FuncPred benchmark, this study shows improvement only on *VWB EG* and *ScreenSpot* with minimal gains compared to the state-of-the-art.)

**Questions:**

See Weaknesses.

---

> ### Author Response · Authors · 2024-11-22
> **Response to Reviewer ZWrk (part 1)**
>
> Thank you for a thorough review that will help us improve the work. Please see below for answers to your questions.
>
> ***
>
> > Q1:Qwen-VL-AutoGUI702k performed worse than SeeClick on VWB EG, RefExp, and ScreenSpot. The paper attributes this
> performance gap to the absence of data. More experiments are needed to support this argument.
>
> A: We agree that a more comprehensive analysis, including the ScreenSpot subsets, would provide a clearer explanation. The performance metrics on these subsets are provided in the table below for reference.
>
> To justify that the gap can be attributed to the **absence of data**, we annotated the trajectories from other sources. As no open-source large-scale trajectory data collected on Apple or desktop software was available, we used the AndroidControl dataset [1] to conduct this experiment. We annotated the trajectories from AndroidControl, expanding the dataset to 772k samples. Fine-tuning Qwen-VL on this augmented dataset led to **improved performance on the ScreenSpot-Mobile and VWB EG subsets** (Row 3). This suggests that dataset expansion can positively impact performance.
>
> Another possible cause is **referring expression format discrepancy**. The RE formats used in the benchmarks differ from our functionality annotations, which could have constrained the benefits of the AutoGUI dataset. For example, our annotations describe functionality like "this element navigates users to the home page," while formats like those in MOTIF and RefExp are more user-intent-oriented, such as "select the text below the images.”
>
> To fill this gap, we reformatted our functionality annotations to align with the user-intent format. For instance, "go to the link element that opens a pop-up window for login." These reformatted samples were incorporated into the AutoGUI-702k dataset for further fine-tuning, resulting in a noticeable improvement on RefExp (Row 4). This demonstrates that aligning the RE format can help reduce the performance gap.
>
> | Model | VWB EG | VWB AG | RefExp | ScreenSpot-Overall | ScreenSpot-Mobile|ScreenSpot-Desktop|ScreenSpot-Web|
> |---|---|---|---|---|---|---|--|
> | seeclick | 39.2 | 27.2 | 58.1 | 53.4 | 65.9 | 52.7 | 45.6 |
> | Qwen-AutoGUI702k | 38.0 | 32.0 | 23.9 | 38.4 | 43.8 | 35.6 | 34.2 |
> | Qwen-AutoGUI702k+AndroidControl | 42.1 | 35.9 | 24.6 | 41.2 | 47.4 | 41.0 | 34.2 |
> | Qwen-VL-Reformatted AutoGUI702k | 40.2 | 33.0 | 31.6 | 40.5 | 49.0 | 35.6 | 34,4 |
>
> In summary, the performance gap can be attributed to both the absence of certain data and discrepancies in referring expression formats. We will revise the experimental analysis in the main paper to incorporate these insights.
>
> ***
>
> > Q2: Is the role of AutoGUI data not as significant as SeeClick data?
>
> A: We’d like to clarify that AutoGUI data is significant compared to the SeeClick-Web portion that is actually annotated by the SeeClick authors.
>
> Only the **web portion** in the SeeClick training data (SeeClick-Web) is generated by the SeeClick authors while the other portions (i.e. Widget captioning, RICOSCA, screen summarization, and LLaVA Pre-train) originate from existing public data sources.
>
> To directly compare the effectiveness of SeeClick-Web and our AutoGUI data, we conducted two experiments: 1) Comparing Qwen-VL fine-tuned on SeeClick-Web versus AutoGUI702k; 2)  Comparing Qwen-VL fine-tuned on the full SeeClick dataset versus a combination of AutoGUI702k and the full SeeClick dataset, excluding SeeClick-Web.
>
> | Models | FuncPred | VWB EG | VWB AG | MOTIF | RefExp | ScreenSpot |
> |---|---|---|---|---|---|---|
> | Qwen-VL w/ SeeClick-Web | 16.9 | 36.7 | 29.1 | 9.9 | 22.1 | 34.8 |
> | Qwen-VL w/ AutoGUI | 43.1 | 38.0 | 32.0 | 15.5 | 23.9 | 38.4 |
> | Qwen-VL w/ SeeClick full | 19.8 | 39.2 | 27.2 | 11.1 | 58.1 | 53.4 |
> | Qwen-VL w/ AutoGUI + SeeClick full w/ SeeClick-Web | 43.8 | 40.0 | 37.9 | 15.7 | 47.3 | 53.6 |
>
> The results show that AutoGUI data outperforms SeeClick-Web **when used independently** (r2 vs. r1). Furthermore, when combined with other data sources, AutoGUI still leads across the four benchmarks. The underperformance of AutoGUI on RefExp is likely due to a format discrepancy between the functionality annotations in AutoGUI and the referring expressions in RefExp.
>
> ***
>
> > Q3: Additional experiment on the Visual appearance and category annotation type.
>
> A: We used GPT-4o-mini to annotate the visual appearance as well as the element category for 125k elements in AutoGUI702k. A description is like: A blue eye icon representing the "Show password" button.
>
> Fine-tuning Qwen-VL with 25k and 125k subsets, we find that the visual appearance & category annotation type is inferior to the proposed functionality annotation type, especially on the FuncPred benchmark.
>
> | Model | FuncPred |RefExp|ScreenSpot|
> |---|---|---|---|
> | QwenVL_25kVisApperance&Category | 4.0 | 7.8 | 9.8 |
> | QwenVL_125kVisApperance&Category | 16.8 | 11.5 | 23.6 |
> | QwenVL_AutoGUI-25k | 21.1 | 10.0 | 16.4 |
> | QwenVL_AutoGUI-125k | 24.6 | 12.7 | 27.0 |

---

> > ### Comment · Reviewer_ZWrk · 2024-11-25
> >
> > Q1:
> >
> > $\textbf{ScreenSpot subsets.}$ In the web scenario, Qwen-AutoGUI702k performs worse than Seeclick on ScreenSpot-Web subset. Could this also be attributed to the absence of mobile and desktop data in the AutoGUI dataset?
> >
> > $\textbf{The absent of data.}$ Qwen-AutoGUI702k+AndroidControl performs better than Qwen-AutoGUI702k on the ScreenSpot-Mobile and VWB EG subsets due to the inclusion of additional Android data. However, the improvement is smaller compared to the gap between Seeclick and Qwen-AutoGUI702k-xxx. Even the Qwen-VL-Reformatted AutoGUI702k, fine-tuned on format-aligned data, performs worse than Seeclick on RefExp and Seeclick benchmark.
> >
> > In summary, the table provided still cannot sufficiently demonstrate the effectiveness of AutoGUI data.
> >
> > Q2:
> >
> > 1. I do know that only the web portion of SeeClick is generated by the SeeClick authors and the others are collected from public. I believe it is neither fair nor reasonable to compare the AutoGUI dataset exclusively to the web portion of the SeeClick training data.
> >
> > $\textbf{Unreasonableness.}$ The Seeclick authors train Qwen-VL on all the SeeClick training dataset to develop SeeClick that is capable of performing the GUI function grounding task. Thus, the full SeeClick training dataset should be treated as a single entity when assessing the effectiveness of a dataset for GUI-related tasks.
> >
> > $\textbf{Unfair.}$ The Seeclick-Web is a GUI appearance grounding dataset that ground the text or icon caption of an element, while the AutoGUI dataset is a function grounding dataset that ground the function of an element. Therefore, comparing $\textit{Qwen-VL w/ AutoGUI}$ with $\textit{Qwen-VL w/ SeeClick-Web}$ on function grounding benchmark (such as FuncPred, RefExp and ScreenSpot) is inherently unfair.
> >
> > 2. The AutoGUI dataset appears less effective than the public portions of SeeClick training data based on the following observations:
> >
> > a) With the inclusion of public portions of SeeClick:
> >
> > - SeeClick demonstrates significant improvements on RefExp and ScreenSpot benchmarks. ($\textit{Qwen-VL w/ SeeClick-Web}$ vs $\textit{Qwen-VL w/ SeeClick full}$)
> >
> > - The AutoGUI model also shows significant improvements on RefExp and ScreenSpot benchmarks. ($\textit{Qwen-VL w/ AutoGUI}$ vs $\textit{Qwen-VL w/ AutoGUI + SeeClick full w/ SeeClick-Web}$)
> >
> > b) With the inclusion of the AutoGUI dataset:
> >
> > - Qwen-VL experiences a notable decline on RefExp and no improvement on ScreenSpot. ($\textit{Qwen-VL w/ SeeClick full}$ vs $\textit{Qwen-VL w/ AutoGUI + SeeClick full w/ SeeClick-Web}$)
> >
> > Q3:
> > No further problems.

---

> ### Author Response · Authors · 2024-11-22
> **Response to Reviewer ZWrk (part 2)**
>
> > Q4: This paper mentions that the ultimate goal of the research is to enable next-generation software automation. However, the work presented here focuses solely on the GUI grounding task, lacking exploration of practical application scenarios.
>
> A: Thank you for your feedback. We would like to clarify that:
>
> - The mention of "software automation" in our paper serves as the broader context for our work. The primary focus of this paper is **GUI grounding**, which we consider a critical capability for vision-language models used in software automation.
>
> - Furthermore, we explicitly state the goal of this research—namely, to provide a dataset rich in functionality semantics to enhance GUI grounding—in the paper title, as well as in the Abstract (lines 27-28) and Introduction (lines 43-44 & 90-94).
>
> We hope these references help clarify the scope of our work and prevent any potential misunderstanding.
>
> ***
>
> > Q5: The experimental results indicate that the proposed AutoGUI dataset has not led to substantial advancements in GUI grounding tasks.
>
> A: Thank you for your interest in the effectiveness of the AutoGUI dataset. We'd like to clarify that
>
> - Qwen-VL fine-tuned with AutoGUI data achieves impressive grounding accuracy on FuncPred, VWB AG, and MOTIF (Table 4).
>
> - Furthermore, the experiment in **Q1** shows that fine-tuned with the AutoGUI dataset augmented by expanding data amount and reformatting, Qwen-VL shows more noticeable improvement. See Q1 for analysis.
>
> - Moreover, the experiment in **Q2** shows that AutoGUI is more significant compared with SeeClick on FuncPred, VWB EG, VWB AG, and MOTIF, although lags on RefExp due to referring expression format discrepancy.
>
> In summary, we believe our AutoGUI dataset can enhance GUI grounding performance and we will continue expanding the data and utilize diverse functionality formats to reduce the gap between AutoGUI data and the benchmarks.
>
> ***
>
> **References:**
>
> [1] On the Effects of Data Scale on Computer Control Agents, NIPS 2024 Datasets & Benchmarks
>
> _We thank Reviewer ZWrk again for the insightful review and feedback and we hope that the above responses adequately address all concerns._

---

> > ### Comment · Reviewer_ZWrk · 2024-11-25
> >
> > Q4:
> >
> > Certainly, the primary focus of this paper is GUI grounding. What I mean is, how do you demonstrate that your contributions to GUI grounding can have a positive impact on software automation as well? Of course, this is not a particularly critical issue.
> >
> > Q5:
> >
> > The utility of AutoGUI data is not yet fully established. Refer to the response to $\textbf{part 1}$ for details.

---

> > > ### Author Response · Authors · 2024-12-04
> > >
> > > Dear Reviewer ZWrk,
> > >
> > > As the discussion phase nears its conclusion, we would like to take this opportunity to express our sincere gratitude for your valuable feedback and constructive engagement throughout this process.
> > >
> > > With only a few hours remaining, we would like to ensure that all your concerns have been thoroughly addressed. In the latest response, we made a dedicated effort to comprehensively address all follow-up questions, including those regarding **Dataset Effectiveness**, **Performance Gap Analysis**, and **SoftWare Automation Applications**. If you have the chance, we would greatly appreciate it if you could review our latest response to confirm whether it fully addresses your concerns.
> > >
> > > We deeply appreciate the time and thoughtfulness you have invested in this discussion and are eager to hear any further thoughts or suggestions you may have. Thank you once again for your support and attention to our work!
> > >
> > > Best regards,
> > >
> > > The Authors

---

### Official Review · Reviewer_PuoX · 2024-11-04

**Soundness:** 3
**Presentation:** 3
**Contribution:** 3
**Rating:** 8
**Confidence:** 4

**Summary:**

This paper develops a new dataset creation pipeline for collecting Graphical User Interface (GUI) element functionality annotations, which they name AutoGUI. It is focused on obtaining high quality annotations of UI interactions in terms functionality descriptions of the different elements in the UI. Ultimately, they focus on obtaining high-quality tuples of (UI Screenshot, UI Element, Functionality). The pipeline first collects UI trajectories by simulating user interactions when interacting with the elements of the UI. Then, each pair of interactions is analyzed by an LLM to identify its functionality (by observing differences between the UI elements and accessibility tree before and after the interaction). They propose a rejection based LLM filtering to discard unsatisfactory functionality annotations. The pipeline mainly focuses on processing websites or Android UI samples. Using the described pipeline, authors use it to collect the AutoGUI-704k dataset. Finally, they finetune a number of VLM baselines on the proposed dataset to show how they improve in the UI grounding and reasoning tasks.

**Strengths:**

- The paper is well written and easy to read.
- The figures presented in the paper are useful for helping understand the presented pipeline with real examples.
- The AutoGUI pipeline offers good advantages over most of its competitors (as shown in Table 1), specially for 1) its scalability and automation (removing the need for costly human annotation), 2) contextualized functionality annotations, 3) dataset size, and 4) coverage of both web and android
- The analysis on data quality comparing different ablations of the method with human annotations helps strengthen the contribution of AutoGUI.
- Sufficient and relevant benchmarks are selected for evaluating the finetuned VLMs in UI grounding tasks.
- Results show that the proposed dataset helps improving GUI grounding performance.

**Weaknesses:**

- The dataset would enjoy more advantages if it contained more platforms other than websites and Android UI. For instance, extending it to other operating systems, and new UI applications.
- The number of baselines appears quite limited. I would like to see the performance of state-of-the-art VLMs (boths open and closed source). Results from closed GPT4o, Claude 3.5 Sonnet, or Gemini would help in comparing methods on the presented benchmarks. Similarly, there are a number of powerful open-source models like Llama3.2 [1], MolMo [2], or larger versions of Qwen2VL, like its 70B variant.
- Authors could have performed more fine tuning experiments. Only QwenVL and Slime models have been finetuned. Finetuning more models would help strengthen the contribution of this dataset. For instance, they provide results on Llava, which they could also finetune.
- We can't see the benefits of the proposed dataset in comparison with other datasets in terms of performance.

[1] https://ai.meta.com/blog/llama-3-2-connect-2024-vision-edge-mobile-devices/
[2] Deitke, Matt, et al. "Molmo and pixmo: Open weights and open data for state-of-the-art multimodal models." arXiv preprint arXiv:2409.17146 (2024).

**Questions:**

- Is it easy to scale AutoGUI to new platforms such as IOS, and computer OS UIs?
- Is it easy to scale AutoGUI to multiple languages?
- Have the authors analyzed the overlap between train and test data to avoid any contamination?
- How does the proposed dataset compare with the other datasets of Table 1 in terms of performance (e.g. benchmark evaluations in Table 4). Are results substantially better in comparison to the same models finetuned on other datasets.
- How does it affect the resolution of the input image when solving this task for VLMs?

---

> ### Author Response · Authors · 2024-11-22
> **Response to Reviewer PuoX (Part 1)**
>
> We thank the reviewer for the insightful comments. Below, we address the concerns:
>
> ***
> >Q1: The dataset would enjoy more advantages if it contained more platforms.
>
> A: We fully agree that expanding our dataset to include more platforms would significantly enhance its diversity and relevance. To address this, we are planning to incorporate GUI data from additional platforms in the next iteration of AutoGUI. Specifically, we intend to collect data on Windows using pywinauto [1], and on iOS using the Xcode Accessibility Inspector [2]. We also hope that the community can extend our annotation pipeline to diverse platforms, further enriching the dataset's utility and applicability.
> ***
> >Q2: The performance of state-of-the-art VLMs (both open and closed source).
>
> A: In response to your concern, we have expanded our evaluation to include **three powerful models**. The results show that GPT-4o, while strong in general tasks, does not perform as well on the GUI element grounding benchmarks. In contrast, Qwen2-VL-72B-Instruct, which has been specifically trained with extensive GUI data, shows superior grounding accuracy over GPT-4o. However, Llama-3.2-11B-Vision-Instruct exhibits suboptimal performance across all benchmarks.
>
> | Models| FuncPred | VWB EG | VWB AG | MOTIF | RefExp | ScreenSpot |
> |------|-----|--------|-----|-------|--------|-------|
> | GPT-4o| 9.8      | 5.6    | 6.8    | 30.5  | 21.8   | 17.8       |
> | Qwen2-VL-72B-Instruct         | 47.7     | 60.5   | 62.1   | 81.3  | 77.7   | 71.4       |
> | Llama-3.2-11B-Vision-Instruct | 4.9      | 7.0    | 3.9    | 19.7  | 5.7    | 11.7       |
> ***
> >Q3: Authors could have performed more fine-tuning experiments.
>
> A: We fine-tuned Llava-1.5-13B with the three scales of AutoGUI data. The results show that this model also **demonstrates improved grounding accuracy across all evaluated benchmarks**, further validating the contribution of our dataset.
>
> | Models         | FuncPred | ScreenSpot | MOTIF | RefExp | VWB EG | VWB AG |
> |--------|----------|------|-------|--------|--------|--------|
> | Llava-1.5-13b (baseline)  | 5.8      | 11.2       | 12.3  | 20.3   | 16.7   | 9.7    |
> | Llava-1.5-13b-AutoGUI25k  | 6.1      | 12.2       | 14.4  | 25.7   | 24.1   | 12.0   |
> | Llava-1.5-13b-AutoGUI125k | 16.0     | 13.4       | 21.2  | 33.3   | 29.2   | 12.6   |
> | Llava-1.5-13b-AutoGUI702k | 45.4     | 15.2       | 25.1  | 35.6   | 29.7   | 21.4   |
>
> ***
> >Q4: The benefits of the AutoGUI dataset compared with other datasets.
>
> A: In response to your concern, we compared AutoGUI702k with two large datasets mentioned in Table 1, i.e., **Wid. Cap.+RICOSCA** (merged due to shared GUI screenshot sources), and **SeeClick-Web**. We did not compare with UI REC/REG and Ferret-UI as the former has been compared in Table 5 while the latter was not open-source.
>
> The table below shows that **the updated AutoGUI dataset (row 4) achieves improved grounding accuracy across all benchmarks**, except RefExp. Upon further analysis, we observed that the referring expression (RE) formats in Wid. Cap.+RICOSCA and RefExp are similar, both incorporating user intents with positional descriptions, which may explain the higher performance on RefExp. For example, RefExp tasks are user intents with position descriptions, such as “click on the third image on the right from the top” (refer to https://huggingface.co/datasets/ivelin/rico_refexp_combined). RICOSCA samples also take on this format (https://huggingface.co/datasets/rootsautomation/RICO-SCA).
>
> Additionally, the RE format of RefExp differs from that of our functionality annotation, which might have limited the perceived benefits of AutoGUI702k. For example, our functionality is phrased as “this element navigates users to the home page”, while RefExp includes user intent-focused REs.
>
> To fill this format-related gap, we have reformatted our functionality annotations to also reflect user intents, such as “go to the link element that opens a pop-up window for log-in”. These reformatted samples were added to the AutoGUI702k dataset. Row 4 shows that **aligning our functionality REs more closely with the RefExp format helps to boost performance on this benchmark**.
>
> | Models                              | FuncPred | VWB EG | VWB AG | MOTIF | RefExp | ScreenSpot |
> |-------------------------------------|----------|--------|--------|-------|--------|------------|
> | Qwen-VL w/ WidgetCaptioning+RICOSCA | 12.6     | 25.7   | 19.4   | 14.7  | 36.0   | 36.5       |
> | Qwen-VL w/ SeeClick-Web             | 16.9     | 36.7   | 29.1   | 9.9   | 22.1   | 34.8       |
> | Qwen-VL w/ AutoGUI702k              | 43.1     | 38.0   | 32.0   | 15.5  | 23.9   | 38.4       |
> | Qwen-VL w/ Reformatted AutoGUI702k  | 50.4     | 40.2   | 33.0   | 20.3  | 31.6   | 40.5       |
>
> In summary, the AutoGUI data demonstrates clear benefits compared with other datasets and we can adjust the functionality annotation format to reduce a potential domain gap.

---

> ### Author Response · Authors · 2024-11-22
> **Response to Reviewer PuoX (Part 2)**
>
> >Q5: Is it easy to scale AutoGUI to new platforms?
>
> A: Yes, it is easy if we can obtain the source code of GUI layouts. For example, one can collect and annotate GUIs on Windows using pywinauto, and iOS using Xcode Accessibility Inspector.
> ***
> >Q6: Is it easy to scale AutoGUI to multiple languages?
>
> A: Yes, scaling AutoGUI to multiple languages is straightforward when **using LLMs with reliable multilingual capabilities**. In our pipeline, we utilize Llama-3-70B, which supports several languages, including Chinese, German, Russian, and Spanish. For example, in the AutoGUI702k dataset, a GUI displayed a SAMSUNG login page in traditional Chinese before interaction, and a Google login page afterward. The LLM translated the Chinese text into English as part of its Chain-of-Thought reasoning, ultimately deducing that the element was used for logging in with Google.
> ***
> >Q7: Have the authors analyzed the overlap between train and test data to avoid any contamination?
>
> A: Yes, we have. Since our focus is on annotating contextual functionality for GUI elements, we define **two elements as distinct if they serve different functions within their respective contexts**. For example, two "magnifier" buttons on the same GUI might have different roles—one for zooming in and the other for searching. To ensure no contamination, we investigated whether the test elements appeared in the training set by checking if bounding box overlapping occurred on the same GUIs. After this analysis, we found no such overlap.
> ***
> >Q8: How does the proposed dataset compare with the other datasets of Table 1 in terms of performance? Are results substantially better in comparison to the same models finetuned on other datasets?
>
> A: The results are better on most of the benchmarks. We kindly ask the reviewer to refer to Q4 for analysis.
> ***
> >Q9: How does it affect the resolution of the input image when solving this task for VLMs?
>
> A: We are sorry that we do not quite understand this question. We guess the reviewer is concerned about **how input image resolution impacts GUI grounding performance**. While this issue is more closely related to model design, we conducted an ablation study to investigate the effect of input image resolution using the SLiME and Qwen2-VL-72B models. Both models employ an image division strategy, where high-resolution screenshots are split into patches and encoded into a visual feature sequence for input to the LLM.
>
> We evaluated three input resolution configurations:
>
> (1) **Vision Encoder’s Native Resolution**: We disabled the division strategy and resized the input screenshot to the size of a single patch (336x336 for SLiME and 224x224 for Qwen2-VL).
>
> (2) **Resized Longest Dimension**: We resized the input screenshot to a longest dimension of 644, maintaining the aspect ratio.
>
> (3) **No Resizing**: The longest dimension of the GUI screenshot, typically greater than 1000, was retained.
>
> The results below show that **input resolution significantly affects GUI grounding performance, with higher resolutions yielding improved accuracy**. This is expected, as GUI content is often dense and complex, requiring models to process fine details for accurate grounding.
>
> | Models | FuncPred | VWB EG | VWB AG | MOTIF | RefExp | ScreenSpot |
> |---|---|---|---|---|---|---|
> | Qwen2-VL-72B-Instruct (Res:224) | 5.4 | 1.9 | 1.9 | 32.1 | 17.5 | 11.5 |
> | Qwen2-VL-72B-Instruct (Res:644) | 28.9 | 31.5 | 21.4 | 63.0 | 58.6 | 45.3 |
> | Qwen2-VL-72B-Instruct (w/o resize) | 47.7 | 60.5 | 62.1 | 81.3 | 77.7 | 71.4 |
> | SLiME-8B-AutoGUI-702k  (Res:336) | 29.7 | 16.2 | 9.6 | 8.7 | 13.5 | 21.0 |
> | SLiME-8B-AutoGUI-702k  (Res:644) | 41.7 | 18.3 | 10.6 | 12.5 | 23.2 | 34.3 |
> | SLiME-8B-AutoGUI-702k (w/o resize) | 62.6 | 25.4 | 13.6 | 20.6 | 26.7 | 44.0 |
> ***
> References:
>
> [1] https://github.com/pywinauto/pywinauto
>
> [2] https://developer.apple.com/documentation/accessibility/accessibility-inspector
> ***
> _We thank the reviewer again for the insightful review and feedback! Hope we have addressed your concerns._

---

> > ### Comment · Reviewer_PuoX · 2024-11-25
> >
> > Thanks to the authors for their responses and for conducting substantial experiments during the rebuttal. Congratulations as well, as the results seem very promising for your method and dataset. I am satisfied with the clarifications provided. Therefore, I am happy to keep my score.

---

> > > ### Author Response · Authors · 2024-11-25
> > > **Appreciation for Positive Feedback**
> > >
> > > We are glad that the clarifications met your expectations. Thank you again for your positive feedback and for recognizing the efforts made in our rebuttal and experiments!

---

### Official Review · Reviewer_dSda · 2024-11-06

**Soundness:** 3
**Presentation:** 3
**Contribution:** 3
**Rating:** 6
**Confidence:** 3

**Summary:**

This paper presents the AutoGUI pipeline for auto-annotating UI element functionality using LLMs, reducing manual work by identifying functions through simulated interaction data. The AutoGUI-704k dataset, constructed by the proposed pipeline, enhances Vision-Language Models in UI understanding. Results show that the dataset improves VLM grounding accuracy, approaching human performance, with automated rejection and verification effectively reducing errors. Human evaluation further demonstrates that the AutoGUI pipeline achieves annotation correctness comparable to trained human annotators.

**Strengths:**

1．AutoGUI pipeline provides a scalable solution to manual UI annotation by using LLMs for functionality labeling, reducing labor and advancing VLM understanding of UI elements.
2.  The pipeline annotates functionality based on UI dynamics, using LLMs to analyze content changes triggered by interactions. This approach enables functionality labeling without manual intervention, capturing detailed functional nuances.
3. AutoGUI-704k dataset covers Web, Mobile device types and UI contexts, valuable for advancing VLM research. The LLM-aided rejection and verification process ensures data quality, reducing manual correction and enhancing annotation reliability.

**Weaknesses:**

1. The experiments focus on specific test sets and benchmarks but lack an analysis of the finetuned model’s generalization across diverse UI types and applications. This may affect the pipeline’s robustness in handling various UI designs, platforms, and complex interactions in real-world settings.

2. Although there are some human checks, the pipeline relies heavily on LLMs for rejection and verification. This raises concerns about whether LLM-based processes alone can consistently maintain high-quality annotations as the dataset scales.

**Questions:**

See the weaknesses.

---

> ### Author Response · Authors · 2024-11-22
> **Response to Reviewer dSda**
>
> Thank you for the insightful review and feedback. Please see below for responses to your concerns.
> ***
>
> >Q1: Any analysis of the fine-tuned model’s generalization across diverse UI types and applications?
>
> A: We calculate the grounding metric values for the **three platforms used in the ScreenSpot** (other benchmarks focus either exclusively on mobile or web scenarios and are not suitable for analysis). See the table below:
>
> | Models               | ScreenSpot Overall | ScreenSpot-mobile (iOS, Android) | ScreenSpot-Desktop (macOS, Windows) | ScreenSpot-Web |
> |----------------------|--------------------|----------------------------------|-------------------------------------|----------------|
> | SeeClick             | 53.4               | 65.9                             | 52.7                                | 45.6           |
> | Qwen-VL-AutoGUI-702k | 38.4               | 43.8                             | 35.6                                | 31.7           |
> | Qwen-VL-AutoGUI702k∗ | 54.2               | 63.5                             | 56.9                                | 41.5           |
> | Slime-AutoGUI-702k   | 44.0               | 50.8                             | 43.4                                | 39.4           |
>
> The fine-tuned models, as well as SeeClick, show lower scores on the desktop and web splits. After examining failure cases, we identified two main factors contributing to this:
>
> **1) Content richness**: The web split contains samples with higher content richness than the mobile split. After downsizing in the preprocessing stage of the models (Qwen receives 448x448 and SLiME receives at most 1344 x 1344), the content becomes more obscure and distorted, likely leading to grounding difficulty.
>
> **2) Data absence**: Collecting desktop data presents additional challenges due to the complexity of installing various software and capturing their UI states.
>
> Despite these challenges, table 4 in the main paper shows that the fine-tuned models still demonstrate **notable scalability** across mobile-related benchmarks (MOTIF and RefExp) and web-related ones (VWB), suggesting a degree of generalization. We believe that the community can leverage our automated annotation pipeline to enrich our dataset, thereby enhancing the model's robustness and performance across various platforms.
> ***
> >Q2. Can LLM-based processes alone consistently maintain high-quality annotations as the dataset scales?
>
> A:  We recognize the importance of maintaining high-quality annotations and would like to clarify that the efficacy of LLM-based annotation and verification has been substantiated in several studies [1-5]. These studies either justify the high quality of LLM-generated annotations or the effectiveness of LLM as a judge.
>
> In our work, the primary GUI data sources include the top 200 most visited domains (Line 946 in Sec. A.2) and commonly used Android built-in apps (Line 966), covering the GUIs frequently encountered in daily use. We develop the AutoGUI annotation pipeline towards these prevailing GUIs and guarantee data quality on these GUI distributions via human verification (Section 4).
>
> However, we acknowledge that **no single annotation method can perfectly generalize across all types of GUI data**. Nevertheless,  the AutoGUI pipeline is designed to be **adaptable**, allowing for the integration of domain-specific rules for annotation, rejection, and verification as needed. We believe the community can continue to extend our pipeline to enhance both scalability and quality.
> ***
> **References:**
>
> [1] CoEvol: Constructing Better Responses for Instruction Finetuning through Multi-Agent Cooperation, EMNLP 2024
>
> [2] Judging LLM-as-a-Judge with MT-Bench and Chatbot Arena, NIPS 2023
>
> [3] TarGEN: Targeted Data Generation with Large Language Models, COLM 2024
>
> [4] Is a large language model a good annotator for event extraction, AAAI 2024
>
> [5] Making large language models better data creator, EMNLP 2023
> ***
> _We thank the reviewer again for the insightful review and feedback! Hope we have addressed your concerns._

---

> ### Author Response · Authors · 2024-11-30
> **Follow-up Comment by the Authors**
>
> Dear Reviewer dSda:
>
> We sincerely thank you for these constructive comments and evaluation of our paper. With the ICLR public discussion phase ending in two days, we kindly ask you to take a look at our responses. Our rebuttal also provided additional experiments in response to your concerns. Please let us know whether our response addresses your concerns or whether there is any further detail we can provide to help address these concerns.
>
> Thank you again for dedicating your time to reviewing our paper.

---

> > ### Comment · Reviewer_dSda · 2024-12-01
> > **Thank you for the rebuttal.**
> >
> > Thank you for your response. It clears some of my concerns. And I would like to keep my positive score unchanged.

---

### Author Response · Authors · 2024-11-23
**Rebuttal by Authors**

The authors would like to thank all reviewers for their appreciation and instructive suggestions!

**Summary**

The authors are encouraged by the following comments:

1. The AutoGUI pipeline reduces the labor and difficulty of GUI functionality annotation (dSda, PuoX).

2. The AutoGUI pipeline ensures annotation reliability and quality (dSda, PuoX, ZWrk, vvNy).

3. The AutoGUI-627k dataset is valuable for advancing VLM research (dSda, vvNy), and offers advantages over most of its competitors (PuoX, vvNy).

4. The paper is well-written and easy to read (PuoX, ZWrk).

5. The figures are useful for understanding the authors' ideas (PuoX).

6. The experiments are comprehensive and sufficient (PuoX).

In our previous response, we addressed concerns raised in the reviews and provided detailed explanations to clarify various aspects of our work. However, to ensure clarity and emphasize the core contributions of our work, we would like to summarize the main points here briefly:

- We build an automatic annotation pipeline that utilizes powerful LLMs (e.g. Llama-3-70B) to generate **_high-quality functionality descriptions_** for massive GUI elements. We believe **this annotation pipeline as well as our 2M training samples** can benefit the community.

- Before submission, our AutoGUI contained 702k training samples. During this discussion phase, we expanded it to a **2M-scale dataset** and witnessed **significant performance gains brought by this augmented dataset** (Please refer to Q1 in the Reply to Reviewer ZWrk's Further Comments (Part 1)).

- We show the promising application of our functionality-focused AutoGUI dataset, which can be used to **boost downstream GUI agent task performance**. Thanks to Reviewer ZWrk's suggestion, we fine-tuned our model with downstream agent task data and observed high success rates. We also show that our model can assist VLM-based planners by providing detailed functionality descriptions of diverse GUIs.

We hope this work supports the community and assists other researchers in scaling our method to larger models and datasets. We will release the annotation pipeline, datasets, and training scripts to facilitate broader adoption and experimentation.


Kind regards,

Authors

---

### Meta-Review · Area_Chair_Wpjr · 2024-12-16

**Metareview:**

AutoGUI introduces a new dataset creation pipeline for generating high-quality GUI element functionality annotations, primarily focusing on websites and Android UIs. The resulting AutoGUI-704k dataset has been employed to fine-tune VLM baselines, leading to improved performance on UI grounding and reasoning tasks.

While the paper presents a valuable contribution to the field, certain concerns remain. A key issue raised by reviewer dSda, even after the rebuttal, is the long-term sustainability of LLM-based processes in maintaining high-quality annotations as the dataset scales. Additionally, reviewer vvNy expressed concerns about the overall utility of AutoGUI and potential data quality issues.

Thanks for flagging ZWrk. This decision has taken into account this.

**Additional Comments On Reviewer Discussion:**

Reviewer vvNy raised concerns about the paper's data quality, particularly regarding the comparison with SeeClick data, and the overall utility of AutoGUI. Additionally, they identified issues with the evaluation in terms of performance drops on RefExp, performance fluctuations, and the lack of clarity surrounding the process for removing invalid samples (including hand-written rules and justification for scoring). While the authors addressed some of these concerns, the reviewer believes the paper would benefit from substantial revisions, particularly regarding data quality and the utility of AutoGUI.

---

### Decision · Program_Chairs · 2025-01-22

Reject